# GRADIENT DESCENT CONVERGES LINEARLY FOR LOGISTIC REGRESSION ON SEPARABLE DATA

## ABSTRACT

We show that running gradient descent on the logistic regression objective guarantees loss $f(\boldsymbol{x}) \le 1.1 \cdot f(\boldsymbol{x}^*) + \varepsilon$, where the error $\varepsilon$ decays exponentially with the number of iterations. This is in contrast to the common intuition that the absence of strong convexity precludes linear convergence of first-order methods, and highlights the importance of variable learning rates for gradient descent. For separable data, our analysis proves that the error between the predictor returned by gradient descent and the hard SVM predictor decays as $\text{poly}(1/t)$, exponentially faster than the previously known bound of $O(\log \log t / \log t)$. Our key observation is a property of the logistic loss that we call multiplicative smoothness and is (surprisingly) little-explored: As the loss decreases, the objective becomes (locally) smoother and therefore the learning rate can increase. Our results also extend to sparse logistic regression, where they lead to an exponential improvement of the sparsity-error tradeoff.

## 1 INTRODUCTION

Logistic regression is one of the most widely used classification methods because of its simplicity, interpretability, and good practical performance. Yet, the convergence behavior of first-order methods on this task is not well understood: In practice gradient descent performs much better than what the theory predicts. In particular, a general analysis of gradient descent for smooth functions implies convergence with the error in function value decaying as $O(1/T)$. Analyses with stronger, linear convergence guarantees generally require the function to satisfy the strong convexity property, which, in contrast to other losses such as the $\ell_2$ loss, the logistic loss only satisfies in a bounded set of solutions around zero. As a result, this introduces an *exponential* runtime dependency on the magnitude of the optimal solution Rätsch et al. (2001); Freund et al. (2018), which is undesirable in practice. This poses a serious obstacle to obtaining favorable error rates for logistic regression that lead to high-precision solutions.

A deeper study into the structure of the exponential and logistic losses was done in Telgarsky & Singer (2012), who showed that, for linearly separable data, greedy coordinate descent achieves linear convergence with a rate that depends on the maximum linear classification margin (i.e. hard SVM margin). Unfortunately, for logistic regression, it also has a $2^m$ dependence on the number of examples, making it inefficient for any real-world task. The significance of the separability of the data for convergence has also been observed in Telgarsky (2013); Freund et al. (2018), who present convergence results based on quantitative measures of separability. Telgarsky (2013) also refines the results of Telgarsky & Singer (2012) for the exponential loss, but still suffers from an exponential overhead originating the multiplicative discrepancy between the exponential and the logistic loss. Interestingly Telgarsky (2013) points out that logistic regression experiments paint a much more favorable picture than the theory predicts. For separable data, Soudry et al. (2018) showed that the gradient descent logistic regression estimator converges to the maximum margin estimator at a rate of $O(\log \log T / \log T)$, which implies function value convergence at a rate of $O(1/T)$. Interestingly, Nacson et al. (2019) experimentally observed that these rates seem to be exponentially improvable if one uses variable step sizes, in the case of logistic regression and shallow neural networks. However, as shown in Ji & Telgarsky (2018), the separability assumption is important, and the $\text{poly}(1/T)$ bound of function value convergence is tight for gradient descent on arbitrary data.

Another approach to obtain high-precision solutions is by using second order methods, which in addition to first order (gradient) information, use second order (Hessian) information about the function. These make use of second order stability properties, such as quasi-self-concordance Bach (2010) combined with Newton's method Karimireddy et al. (2018), or ball oracles Carmon et al. (2020); Adil et al. (2021). Such approaches are generally not suitable for large-scale applications because of their reliance on repeated calls to large linear system solvers.

**Our work.** In this paper, we show that (under appropriate assumptions) we can get the best of both worlds of first and second order methods, thus giving a partial explanation for the excellent performance that first-order methods have on logistic regression in practice. In particular, given a binary classification instance ($\boldsymbol{A} \in \{-1,1\}^{m \times n}$, $\boldsymbol{b} \in \{-1,1\}^m$) with associated logistic loss $f(\boldsymbol{x}) = \sum_i \log(1 + \exp(-b_i(\boldsymbol{A}\boldsymbol{x})_i))$, we show that simple variants of gradient descent return a solution with $f(\boldsymbol{x}) \le (1 + \delta) \cdot f(\boldsymbol{x}^*) + \varepsilon$ after $O\left(K\left(\frac{1}{\delta} + \log \frac{f(\boldsymbol{0})}{\varepsilon}\right)\right)$ iterations, where $K = \text{poly}(n, \|\boldsymbol{x}^*\|)$. Even though the error still decays as $1/T$ in the worst case because of the $\frac{1}{\delta}$ dependence, the additive error is now $\delta f(\boldsymbol{x}^*)$ instead of $\delta f(\boldsymbol{0})$, allowing for much faster convergence when the optimal loss $f(\boldsymbol{x}^*)$ is smaller (which is our measure of linear separability of the data). For linearly separable data, i.e. as $f(\boldsymbol{x}^*)$ approaches $0$, the convergence becomes linear. We also show that the distance to the maximum margin estimator $\left\|\frac{\boldsymbol{x}}{\|\boldsymbol{x}\|_2} - \frac{\boldsymbol{x}^*}{\|\boldsymbol{x}^*\|_2}\right\|_2$ decays as $1/T$, exponentially improving over the $\log \log T / \log T$ bound of Soudry et al. (2018).

Instead of properties like Lipschitzness, smoothness, strong convexity that are commonly used in the study of first order methods, we find that there are two properties that are more relevant to the structure of the logistic regression problem. The first one is *second order robustness*, which means that the Hessian is stable (in a spectral sense) in any small enough norm ball Cohen et al. (2017). This is closely related to quasi-self-concordance, a property that has been previously used in the analysis of second order algorithms Bach (2010). The second property is what we call *multiplicative smoothness*, which means that the function is locally smooth, with the smoothness constant being proportional to the function value (loss). Together, these properties show that, as the loss decreases, the objective becomes (locally) smoother and therefore the learning rate can increase. This motivates a variable step size schedule that is inversely proportional to the loss, thus making larger steps as the solution approaches optimality. This in fact agrees with the observations of Soudry et al. (2018); Nacson et al. (2019) on the importance of a variable learning rate. As can be seen in the toy example from Soudry et al. (2018) in Figure 1, simply replacing the fixed learning rate $\eta$ used in Soudry et al. (2018) by an increasing learning rate $\eta \cdot f(\boldsymbol{x}^0)/f(\boldsymbol{x}^T)$ yields an exponential improvement, both in loss and distance to the maximum margin estimator.

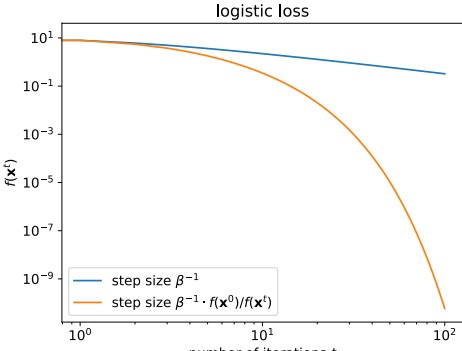 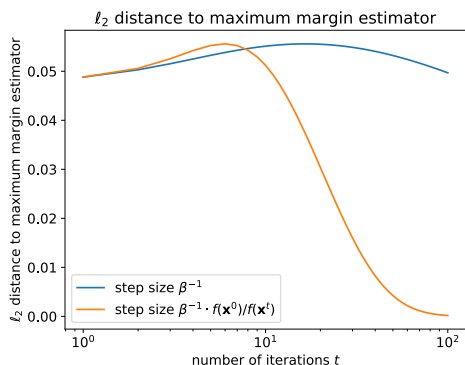

Figure 1: Comparison between fixed and increasing step sizes in the toy example from Figure 1 of Soudry et al. (2018). The fixed step size is set to $\beta^{-1} := \|\boldsymbol{A}\|_2^{-2}$, and the increasing to $\beta^{-1} f(\boldsymbol{x}^0)/f(\boldsymbol{x}^T)$. The estimator error is defined as $\|\boldsymbol{x}^t/\|\boldsymbol{x}^t\|_2 - \boldsymbol{x}^*/\|\boldsymbol{x}^*\|_2\|_2$.

| Algorithm | Order | Guarantee | Runtime Error Dependence |
|---|---|---|---|
| Gradient descent | First | $f(\boldsymbol{x}) \leq f(\boldsymbol{x}^*) + \varepsilon$ | $m/\varepsilon$ |
| Accelerated gradient descent | First | $f(\boldsymbol{x}) \leq f(\boldsymbol{x}^*) + \varepsilon$ | $\sqrt{m/\varepsilon}$ |
| Newton/Trust region | Second | $f(\boldsymbol{x}) \leq f(\boldsymbol{x}^*) + \varepsilon$ | $\log(m/\varepsilon)$ |
| This paper | First | $f(\boldsymbol{x}) \leq (1 + \delta) \cdot f(\boldsymbol{x}^*) + \varepsilon$ | $\delta^{-1} + \log(m/\varepsilon)$ |

Table 1: Algorithms for logistic regression and dependence on $m/\varepsilon$ (omitting extra $\mathrm{polylog}(m, n)$ factors). Algorithms with exponential dependences on any problem parameter are ommitted.

| Algorithm | Guarantee | Sparsity | Order |
|---|---|---|---|
| Shalev-Shwartz et al. (2010) | $f(\boldsymbol{x}) \leq f(\boldsymbol{x}^*) + \varepsilon$ | $\|\boldsymbol{x}^*\|_1^2 \, m/\varepsilon$ | First |
| This paper | $f(\boldsymbol{x}) \leq (1 + \delta) \cdot f(\boldsymbol{x}^*) + \varepsilon$ | $\|\boldsymbol{x}^*\|_1^2 \left(\delta^{-1} + \log(m/\varepsilon)\right)$ | First |

Table 2: Algorithms for sparse logistic regression

## 1.1 Sparse logistic regression

In practice, it is often important to force the solution of a logistic regression problem to be *sparse*, i.e. have only a few non-zero entries, which is a form of feature selection. This is because most of the features might only be marginally useful, and thus one can drastically reduce the size of the model while not significantly sacrificing the predictive performance. Apart from computational efficiency, feature selection is also important to improve interpretability and avoid overfitting.

Most progress in sparse optimization has focused on objective functions with condition number bounded by some $\kappa > 0$. Results in this line of work guarantee a solution with relaxed sparsity $s' \geq s$, where $s$ is the target sparsity, and algorithms include lasso, orthogonal matching pursuit (OMP), and iterative hard thresholding (IHT) Natarajan (1995); Blumensath & Davies (2009); Shalev-Shwartz et al. (2010); Jain et al. (2011; 2014); Axiotis & Sviridenko (2021; 2022). The state of the art result by Axiotis & Sviridenko (2022) gives a sparsity of $s' = O(\kappa) \cdot s$ using a variant of the IHT algorithm.

However, the condition number of the logistic loss is unbounded, because it is not strongly convex. Therefore, these results do not directly apply, although they do apply to $\ell_2$-regularized logistic regression. Some works Van de Geer (2008); Bunea (2008) have analyzed lasso methods for logistic regression without condition number assumptions, and Shalev-Shwartz et al. (2010) provides three different analyses for smooth but not strongly convex functions. These apply to logistic regression and give a sparsity of $O\left(\|\boldsymbol{x}^*\|_1^2 \frac{m}{\varepsilon}\right)$ to achieve a loss of $f(\boldsymbol{x}) \leq f(\boldsymbol{x}^*) + \varepsilon$. The most practical of these is a forward greedy selection algorithm, which is also known as greedy coordinate descent.

**Our work.** Using the second order stability and multiplicative smoothness properties, we show that a slight variation of greedy coordinate descent gives a sparsity of

$$O\left(\|\boldsymbol{x}^*\|_1^2 \left(\delta^{-1} + \log(m/\varepsilon)\right)\right)$$

and a loss of $f(\boldsymbol{x}) \leq (1 + \delta) \cdot f(\boldsymbol{x}^*) + \varepsilon$. As long as the $1 + \delta$ approximation in front of $f(\boldsymbol{x}^*)$ is tolerated, as is the case when $f(\boldsymbol{x}^*) \ll m$, this implies an exponential improvement in the $\varepsilon$ dependence from $\frac{m}{\varepsilon}$ to $\log \frac{m}{\varepsilon}$. In addition, our analysis does not require (but is also not affected by) fully corrective steps, in which the function is fully re-optimized over the support of the current solution.

## 2 Preliminaries

**Notation.** We denote $[n] = \{1, 2, \ldots, n\}$. We will use **bold** to refer to vectors or matrices. We denote by $\boldsymbol{0}$ the all-zero vector, $\boldsymbol{1}$ the all-one vector, $\boldsymbol{O}$ the all-zero matrix, and by $\boldsymbol{I}$ the identity

matrix (with dimensions understood from the context). Additionally, we will denote by $\mathbf{1}_i$ the $i$-th basis vector, i.e. the vector that is 0 everywhere except at position $i$.

In order to ease notation and where not ambiguous for two vectors $\boldsymbol{x}, \boldsymbol{y} \in \mathbb{R}^n$, we denote by $\boldsymbol{x}\boldsymbol{y} \in \mathbb{R}^n$ a vector with elements $(\boldsymbol{x}\boldsymbol{y})_i = x_i y_i$, i.e. the element-wise multiplication of two vectors $\boldsymbol{x}$ and $\boldsymbol{y}$. In contrast, we denote their inner product by $\langle \boldsymbol{x}, \boldsymbol{y} \rangle$ or $\boldsymbol{x}^\top \boldsymbol{y}$. Similarly, $\boldsymbol{x}^2 \in \mathbb{R}^n$ will be the element-wise square of vector $\boldsymbol{x}$.

For any vector $\boldsymbol{x} \in \mathbb{R}^n$ and set $S \subseteq [n]$, we denote by $\boldsymbol{x}_S$ the vector that results from $\boldsymbol{x}$ after zeroing out all the entries except those in positions given by indices in $S$. We will also use the notation $\nabla_S f(\boldsymbol{x}) := (\nabla f(\boldsymbol{x}))_S$ to denote the restriction of a gradient to $S$.

We use the notation $\widetilde{O}(\cdot)$ to hide $\operatorname{poly}\log(n, m)$ factors in $O$-notation, where $n$ is the dimension of the problem and $m$ is the number of examples.

**Norms.** For any $p \in (0, \infty)$ and weight vector $\boldsymbol{w} \geq \mathbf{0}$, we define the weighted $\ell_p$ norm of a vector $\boldsymbol{x} \in \mathbb{R}^n$ as:

$$\|\boldsymbol{x}\|_{p, \boldsymbol{w}} = \left( \sum_i w_i x_i^p \right)^{1/p}.$$

For $p = 0$, we denote $\|\boldsymbol{x}\|_0 = |\{i \mid x_i \neq 0\}|$ to be the *sparsity* of $\boldsymbol{x}$. For $p = \infty$, we denote $\|\boldsymbol{x}\|_\infty = \max_i |x_i|$ to be the maximum absolute value of $\boldsymbol{x}$.

**Smoothness and convexity.** A differentiable function $f : \mathbb{R}^n \to \mathbb{R}$ is called *convex* if for any $\boldsymbol{x}, \boldsymbol{y} \in \mathbb{R}^n$ we have $f(\boldsymbol{y}) \geq f(\boldsymbol{x}) + \langle \nabla f(\boldsymbol{x}), \boldsymbol{y} - \boldsymbol{x} \rangle$. Furthermore, $f$ is called $\beta$-*smooth (with respect to some norm $\|\cdot\|$)* for some real number $\beta > 0$ if for any $\boldsymbol{x}, \boldsymbol{y} \in \mathbb{R}^n$ we have $f(\boldsymbol{y}) \leq f(\boldsymbol{x}) + \langle \nabla f(\boldsymbol{x}), \boldsymbol{y} - \boldsymbol{x} \rangle + (\beta/2) \|\boldsymbol{y} - \boldsymbol{x}\|^2$. If $f$ is only $\beta$-smooth along $s$-sparse directions (i.e. only for $\boldsymbol{x}, \boldsymbol{y} \in \mathbb{R}^n$ such that $\|\boldsymbol{y} - \boldsymbol{x}\|_0 \leq s$), then we call $f$ $\beta$-smooth *at sparsity level $s$* and denote the smallest such $\beta$ by $\beta_s$ and call it the *restricted smoothness constant* (at sparsity level $s$).

## 3   LOGISTIC REGRESSION ANALYSIS VIA MULTIPLICATIVE SMOOTHNESS

In the logistic regression problem, our goal is to minimize the function $f(\boldsymbol{x}) = \sum\limits_{i=1}^{m} \log(1 + e^{-(\boldsymbol{A}\boldsymbol{x})_i})$, where $\boldsymbol{A} \in \mathbb{R}^{m \times n}$ is a data matrix[1]

Our starting point, as is usually the case with first-order methods, will be the second order Taylor expansion of $f$:

$$f(\boldsymbol{x} + \widetilde{\boldsymbol{x}}) = f(\boldsymbol{x}) + \langle \nabla f(\boldsymbol{x}), \widetilde{\boldsymbol{x}} \rangle + \frac{1}{2} \langle \widetilde{\boldsymbol{x}}, \nabla^2 f(\bar{\boldsymbol{x}})\widetilde{\boldsymbol{x}} \rangle, \tag{1}$$

where, by the mean value theorem for twice continuously differentiable functions, $\bar{\boldsymbol{x}}$ is entry-wise between $\boldsymbol{x}$ and $\boldsymbol{x}'$, and $\nabla^2 f(\bar{\boldsymbol{x}})$ is the Hessian of $f$ at $\bar{\boldsymbol{x}}$. In fact, as long as the step $\widetilde{\boldsymbol{x}}$ is not too large, the Hessian at $\bar{\boldsymbol{x}}$ will not differ much (spectrally) from the Hessian at $\boldsymbol{x}$. This is because of the following property of the logistic function called *second order robustness* Cohen et al. (2017), which is also very closely related to quasi-self-concordance Bach (2010).

**Definition 3.1** (Second-order robustness). *A twice differentiable function $f : \mathbb{R}^n \to \mathbb{R}$ is called $q$-second order robust with respect to a norm $\|\cdot\|$ if its Hessian is stable in any $(1/q)$-sized $\|\cdot\|$-ball, i.e. for any $\boldsymbol{x}, \boldsymbol{x}' \in \mathbb{R}^n$ such that $\|\boldsymbol{x}' - \boldsymbol{x}\| \leq 1/q$, we have $\frac{1}{2}\nabla^2 f(\boldsymbol{x}) \preceq \nabla^2 f(\boldsymbol{x}') \preceq 2\nabla^2 f(\boldsymbol{x})$.*

It is not hard to see that $f$ is $2M$-second order robust with respect to the $\ell_1$ norm, where $M$ is a upper bound on the entries of $\boldsymbol{A}$ in absolute value. Because of this, (1) implies the much simpler

$$f(\boldsymbol{x} + \widetilde{\boldsymbol{x}}) = f(\boldsymbol{x}) + \langle \nabla f(\boldsymbol{x}), \widetilde{\boldsymbol{x}} \rangle + \langle \widetilde{\boldsymbol{x}}, \nabla^2 f(\boldsymbol{x})\widetilde{\boldsymbol{x}} \rangle, \tag{2}$$

as long as $\|\widetilde{\boldsymbol{x}}\|_1 \leq 1/(2M)$. We can easily calculate that $\nabla f(\boldsymbol{x}) = -\boldsymbol{A}^\top (\mathbf{1} - \sigma(\boldsymbol{A}\boldsymbol{x}))$, where $\sigma(t) = 1/(1 + e^{-t})$ is the sigmoid function, and $\nabla^2 f(\boldsymbol{x}) = \boldsymbol{A}^\top \operatorname{diag}(\boldsymbol{w}(\boldsymbol{x}))\boldsymbol{A}$, where $\boldsymbol{w}(\boldsymbol{x}) =$

---

[1]This formulation is without loss of generality, because we can incorporate the binary $\pm 1$ labels into the matrix $\boldsymbol{A}$ and assume that all the labels are positive.

$\sigma(\boldsymbol{x})(\boldsymbol{1} - \sigma(\boldsymbol{x}))$ are diagonal weights. Now, we should note that the second order term of (1) can be re-written as $\frac{1}{2}\langle \boldsymbol{w}(\boldsymbol{x}), (\boldsymbol{A}\widetilde{\boldsymbol{x}})^2 \rangle$. This term, whose magnitude is what will determine the step size of the algorithm and in turn the bound on the total number of iterations, becomes smaller as the weights $\boldsymbol{w}(\boldsymbol{x})$ become smaller. The crucial observation is that these weights are bounded in a way that depends on the *loss* of $\bar{\boldsymbol{x}}$, concretely:

$$\sum_{i=1}^{m} (\boldsymbol{w}(\boldsymbol{x}))_i \leq f(\boldsymbol{x}) \, . \tag{3}$$

In other words, as the loss decreases, $f$ becomes *smoother* (in an appropriate sense). This is the main observation on which our analysis is based, and is what allows the algorithm to employ a step size that is *inversely proportional* to the loss.

**Multiplicative smoothness.** The above discussion motivates the following definition of *multiplicative smoothness*. This is related to the usual definition of smoothness but also incorporates the property that the function becomes smoother as the loss decreases.

**Definition 3.2** (Multiplicative smoothness). *We call a twice differentiable function $f : \mathbb{R}^n \to \mathbb{R}_{>0}$ $\mu$-multiplicatively smooth with respect to a norm $\|\cdot\|$, if for any $\boldsymbol{x}, \widetilde{\boldsymbol{x}} \in \mathbb{R}^n$ we have*

$$\frac{\widetilde{\boldsymbol{x}}^{\top} \nabla^2 f(\boldsymbol{x}) \widetilde{\boldsymbol{x}}}{f(\boldsymbol{x})} \leq \mu \|\widetilde{\boldsymbol{x}}\|^2 \, .$$

Our use of a general norm is not an over-generalization, since as we will see the $\ell_1$ norm is more suitable for sparse logistic regression, and the $\ell_2$ norm is more suitable for the unrestricted case. In fact, it can be proved that $f$ is $M^2$-multiplicatively smooth with respect to the $\ell_1$ norm, where we remind that $M$ is a bound on the entries of $\boldsymbol{A}$ in absolute value.

In the following sections, we will see how the second order robustness and multiplicative smoothness properties play into the design and analysis of algorithms for sparse and general logistic regression.

## 4 SPARSE LOGISTIC REGRESSION

As we saw, the logistic loss is $2M$-second order robust and $M^2$-multiplicatively smooth with respect to the $\ell_1$ norm. This is an ideal norm for *sparse* logistic regression, where in addition to minimizing the loss we want to restrict the solution to have few non-zero entries. In particular, it yields a variant of the $\ell_1$ gradient descent algorithm (aka greedy coordinate descent), which is presented in Algorithm 1.

---

**Algorithm 1** Greedy Coordinate Descent

---

1: **procedure** GREEDYCOORDINATEDESCENT($\boldsymbol{x}^0, T, M, B$)

2:      Let $f(\boldsymbol{x}) := \sum\limits_{i=1}^{m} \log(1 + e^{-b_i(\boldsymbol{A}\boldsymbol{x})_i})$

3:      **for** $t = 0, \dots, T-1$ **do**

4:          For all $i \in [n]$ define $\zeta_i = \begin{cases} \lambda_t & \text{if } x_i^t = 0 \\ 0 & \text{if } \|\boldsymbol{x}^t\|_1 \geq B \text{ and } \nabla_i f(\boldsymbol{x}^t) \cdot x_i^t < 0 \\ 1 & \text{otherwise} \end{cases}$

5:          $i \leftarrow \operatorname{argmax}_i \{\zeta_i |\nabla_i f(\boldsymbol{x}^t)|\}$

6:          $\eta \leftarrow (2M \max\{M f(\boldsymbol{x}^t), |\nabla_i f(\boldsymbol{x}^t)|\})^{-1}$

7:          $x_i^{t+1} \leftarrow x_i^t - \eta \nabla_i f(\boldsymbol{x}^t)$

     **return** $\boldsymbol{x}^T$

---

The first thing that should be noted about this algorithm is the crucial parameters $\lambda_t$. These parameters offer a quantitative threshold between sparsity and speed of convergence. In particular, when $\lambda_t$ is 1, then all entries (regardless of whether they are zero or not) are treated the same. When $\lambda_t \ll 1$, on the other hand, the gradient entries corresponding to zero entries are discounted by a factor $\ll 1$, thus making the algorithm less eager to update these as opposed to non-zero entries, whose update doesn't increase sparsity.

A practical consideration about Algorithm 1 is order. The second condition in line 4 is to make sure that the $\ell_1$ norm of the solution never exceeds a given bound on the $\ell_1$ norm of the optimal solution. This check is useful for the theoretical analysis but should likely be removed in any practical implementation.

We are ready for the main theorem of this section. In the proof, which can be found in Appendix A.2.2, we present an analysis of Algorithm 1 for sparse logistic regression. In addition to an upper bound $B \geq \|\boldsymbol{x}^*\|_\infty$, it also requires an approximation $B_1$ of $\|\boldsymbol{x}^*\|_1$. One possible approach is to approximate it by $B$, but in practice this would be a learning rate hyperparameter to be tuned.

**Theorem 4.1** (Sparse logistic regression). *Given a binary classification instance* $(\boldsymbol{A} \in [-M, M]^{m \times n}, \boldsymbol{b} \in \{1, -1\}^m)$ *and for any solution* $\boldsymbol{x}^* \in [-B, B]^n$ *with* $M \geq \max\{\|\boldsymbol{x}^*\|_\infty^{-1}, B^{-1}\}$ [2] *and a known parameter* $B_1 \in \left[\frac{1}{C}\|\boldsymbol{x}^*\|_1, \|\boldsymbol{x}^*\|_1\right]$ *for some* $C \geq 1$, *Algorithm 1 with* $\lambda_t = \min\{B_1/\|\boldsymbol{x}^t\|_1, 1\}$, *initial solution* $\boldsymbol{x}^0 \in \mathbb{R}^n$, *and error tolerance* $0 < \varepsilon < m/2$ *returns a solution* $\boldsymbol{x}$ *with*

$$f(\boldsymbol{x}) \leq (1 + \delta) \cdot f(\boldsymbol{x}^*) + \varepsilon$$

*and sparsity*

$$s' := \|\boldsymbol{x}\|_0 = O\left(\|\boldsymbol{x}^*\|_1^2 M^2 \left(\frac{1}{\delta} + \log \frac{f(\boldsymbol{x}^0) - f(\boldsymbol{x}^*)}{\varepsilon}\right)\right)$$

*in*

$$T = O\left(\left(\|\boldsymbol{x}\|_0^2 + \|\boldsymbol{x}^*\|_0^2\right) M^2 B^2 C^2 \left(\frac{1}{\delta} + \log \frac{f(\boldsymbol{x}^0) - f(\boldsymbol{x}^*)}{\varepsilon}\right)\right)$$

*iterations, for any choice of* $\delta \in (0, 1)$ *and parameter* $c > 0$. *Each iteration consists of evaluating the logistic regression gradient* $\nabla f$ *plus* $O(m + n)$ *additional time.*

**Corollary 4.2.** *If* $M, B, C \leq \widetilde{O}(1)$ *and* $\boldsymbol{x}^*$ *is s-sparse, then Algorithm 1 with* $\lambda_t = \min\{1/\|\boldsymbol{x}^t\|_1, 1\}$ *returns a solution* $\boldsymbol{x}$ *with*

$$f(\boldsymbol{x}) \leq 1.1 \cdot f(\boldsymbol{x}^*) + \varepsilon$$

*and sparsity*

$$s' := \|\boldsymbol{x}\|_0 = \widetilde{O}\left(s^2 \log \frac{1}{\varepsilon}\right)$$

*in*

$$T = \widetilde{O}\left(s^4 \log^3 \frac{1}{\varepsilon}\right)$$

*iterations.*

It is useful to compare these results to the results of Shalev-Shwartz et al. (2010) for sparse optimization of general smooth convex functions. Even though they achieve the stronger error bound of $f(\boldsymbol{x}) \leq f(\boldsymbol{x}^*) + \varepsilon$, the sparsity of the final solution is in the order of $s^2 \frac{m}{\varepsilon}$, which has an exponentially worse error dependence than $s^2 \log \frac{m}{\varepsilon}$. Therefore, if the approximation rate $(1 + \delta)$ is tolerable in front of $f(\boldsymbol{x}^*)$, then one can obtain exponentially faster sparsity and convergence.

If we are willing to perform fully corrective steps as described in Algorithm 2, then we can get a cleaner and slightly simpler analysis. This is presented in Theorem 4.3 and proved in Appendix A.2.3. Fully corrective steps can be useful when there is an efficient (dense) optimization algorithm and one wishes to use it as a black box for sparse optimization. In practice, one does not need to perform a full correction, but only a small number of corrective (usually gradient) steps over the current support of the solution.

---

[2] the theorem can be stated without this additional constraint, but we include it because it makes the bounds considerably simpler

---

**Algorithm 2** Greedy coordinate descent with fully corrective steps

---

1: **procedure** FULLYCORRECTIVEGREEDYCOORDINATEDESCENT($\boldsymbol{x}^0, T, M, B$)
2:     Let $f(\boldsymbol{x}) := \sum_{i=1}^{m} \log(1 + e^{-b_i(\boldsymbol{A}\boldsymbol{x})_i})$
3:     $S^0 \leftarrow \text{supp}(\boldsymbol{x}^0)$
4:     **for** $t = 0, \ldots, T - 1$ **do**
5:         $i \leftarrow \text{argmax}_i \{|\nabla_i f(\boldsymbol{x}^t)|\}$
6:         $S^{t+1} \leftarrow S^t \cup \{i\}$
7:         $\boldsymbol{x}^{t+1} \leftarrow \underset{\boldsymbol{x}:\text{supp}(\boldsymbol{x})\subseteq S^{t+1}}{\text{argmin}} f(\boldsymbol{x})$
        **return** $\boldsymbol{x}^T$

---

**Theorem 4.3** (Sparse logistic regression with fully corrective steps). *Given a binary classification instance* ($\boldsymbol{A} \in [-M, M]^{m \times n}, \boldsymbol{b} \in \{1, -1\}^m$) *and for any solution* $\boldsymbol{x}^* \in \mathbb{R}^n$*, Algorithm 2 with error tolerance* $0 < \varepsilon < m/2$ *and initial solution* $\boldsymbol{x}^0$ *returns a solution* $\boldsymbol{x}$ *with*

$$f(\boldsymbol{x}) \leq (1 + \delta) \cdot f(\boldsymbol{x}^*) + \varepsilon$$

*and sparsity*

$$s' := \|\boldsymbol{x}\|_0 = \|\boldsymbol{x}^0\|_0 + O\left(\|\boldsymbol{x}^*\|_1^2 M^2 \left(\frac{1}{\delta} + \log \frac{f(\boldsymbol{x}^0) - f(\boldsymbol{x}^*)}{\varepsilon}\right)\right)$$

*in* $T = \|\boldsymbol{x}\|_0$ *iterations, for any choice of* $\delta \in (0, 1)$*. Each iteration consists of evaluating the logistic regression gradient* $\nabla f$*, solving a logistic regression problem on* $s'$ *variables, plus* $O(m+n)$ *additional time.*

## 5 DENSE LOGISTIC REGRESSION

In this section, our goal is to minimize the logistic function $f$ without any constraint on the sparsity of the solution. The results of the Section 4 applied to a full sparsity of $n$ already imply Corollary 5.1.

**Corollary 5.1** (Dense logistic regression). *Given a binary classification instance* ($\boldsymbol{A} \in [-M, M]^{m \times n}, \boldsymbol{b} \in \{-1, 1\}^m$) *and for any solution* $\boldsymbol{x}^* \in [-B, B]^n$ *with* $M \geq \max\left\{\|\boldsymbol{x}^*\|_\infty^{-1}, B^{-1}\right\}$*, Algorithm 1 with* $\lambda_t = 1$ *for all t, initial solution* $\boldsymbol{x}^0 \in \mathbb{R}^n$*, and error tolerance* $0 < \varepsilon < m/2$ *returns a solution* $\boldsymbol{x}$ *with*

$$f(\boldsymbol{x}) \leq (1 + \delta) \cdot f(\boldsymbol{x}^*) + \varepsilon$$

*in*

$$T = O\left(n^2 M^2 B^2 \left(\frac{1}{\delta} + \log \frac{f(\boldsymbol{x}^0) - f(\boldsymbol{x}^*)}{\varepsilon}\right)\right).$$

*iterations, for any choice of* $\delta \in (0, 1)$*. Additionally,* $\|\boldsymbol{x}\|_\infty \leq B + \frac{1}{2M}$*. Each iteration consists of evaluating the logistic regression gradient* $\nabla f$ *plus* $O(m + n)$ *additional time.*

Even though Corollary 5.1 has the same favorable convergence in terms of $\delta$ and $\varepsilon$ as Theorem 4.1, based on practical intuitions we would expect ($\ell_2$-based) gradient descent to perform better than greedy coordinate descent, which only updates one coordinate at a time, while having access to the full gradient. In fact, we can verify that the logistic loss does have the multiplicative smoothness condition with respect to the $\ell_2$ norm, albeit in an almost trivial sense:

$$\langle \boldsymbol{w}(\boldsymbol{x}), (\boldsymbol{A}\boldsymbol{x})^2 \rangle \leq \|\boldsymbol{w}(\boldsymbol{x})\|_1 \|\boldsymbol{A}\boldsymbol{x}\|_\infty^2 \leq f(\boldsymbol{x}) \|\boldsymbol{A}\|_{2\to\infty}^2 \|\boldsymbol{x}\|_2^2 \leq f(\boldsymbol{x})\beta \|\boldsymbol{x}\|_2^2 .$$

Here, using the inequality $\|\boldsymbol{A}\|_{2\to\infty}^2 \leq \|\boldsymbol{A}\|_2^2 := \beta$ implies $\beta$-multiplicative smoothness with respect to the $\ell_2$ norm. Unfortunately, this is not significantly better than the $\ell_1$ case: The number of iterations will be proportional to $\beta \|\boldsymbol{x}^*\|_2^2$, which can be $\gg m$.

Table 3: Upper bounds on the quantity $\left\langle \boldsymbol{w}(\boldsymbol{x}), (\boldsymbol{A}\nabla f(\boldsymbol{x}))^2 \right\rangle / \left( f(\boldsymbol{x})m^{-1} \|\boldsymbol{A}\nabla f(\boldsymbol{x})\|_2^2 \right)$. Shown here is the maximum of this over $\boldsymbol{x}$ being one one of the first 1000 iterates.

| Dataset | Max ratio | Dataset | Max ratio | Dataset | Max ratio |
|---------|-----------|---------|-----------|---------|-----------|
| letter | 0.40 | skin | 0.44 | census | 0.50 |
| rcv1.test | 0.36 | w8all | 0.40 | adult | 0.40 |
| ijcnn1 | 0.47 | shuttle.binary | 0.37 | poker | 0.36 |
| vehv2binary | 0.37 | kddcup04.phy | 0.36 | nomao | 0.50 |
| magic04 | 0.37 | kddcup04.bio | 0.48 | covtype | 0.36 |

Interestingly, real logistic regression instances exhibit the $\ell_2$ multiplicative smoothness property with significantly better constants. In our experiments we found that along the path of gradients encountered by gradient descent in a variety of instances, the following property was true:

$$\left\langle \boldsymbol{w}(\boldsymbol{x}), (\boldsymbol{A}\nabla f(\boldsymbol{x}))^2 \le f(\boldsymbol{x})\beta m^{-1} \|\nabla f(\boldsymbol{x})\|_2^2 \right.$$

This is an *effective $\beta m^{-1}$-multiplicative smoothness property*, because it is only assumed to be true for $\boldsymbol{x}$'s encountered by the gradient descent algorithm. As such, it is an empirical property. In order to check our hypothesis, we have run the gradient descent algorithm with the step sizes that are implied by Theorem 5.2, which we will see later. For each of the 15 experiments, we have run gradient descent for 1000 iterations, and calculated the maximum of the following quantity, over all iterations:

$$\frac{\left\langle \boldsymbol{w}(\boldsymbol{x}), (\boldsymbol{A}\nabla f(\boldsymbol{x}))^2 \right\rangle}{f(\boldsymbol{x})m^{-1} \|\boldsymbol{A}\nabla f(\boldsymbol{x})\|_2^2}.$$

If this is bounded by 1, and using the fact that $\|\boldsymbol{A}\nabla f(\boldsymbol{x})\|_2^2 \le \beta \|\nabla f(\boldsymbol{x})\|_2^2$, this implies that $f$ is effectively $\beta m^{-1}$-multiplicatively smooth with respect to the $\ell_2$ norm. Indeed, as we can see in Table 3, these values are indeed less than 1 for all datasets and all iterations.

In the following, our plan is to prove convergence, *assuming* that $f$ has the multiplicative smoothness property with the constants in our hypothesis above. Under this assumption, we can now prove a much stronger convergence theorem (here we are also using the fact that $M^2 \le \beta$ to replace $2M$- by $2\sqrt{\beta}$-second order robustness):

**Theorem 5.2.** *Let $f : \mathbb{R}^n \to \mathbb{R}$ be a convex function that is $2\sqrt{\beta}$-second order robust with respect to the $\ell_1$ norm and $\beta m^{-1}$-multiplicatively smooth with respect to the $\ell_2$ norm. Let $\boldsymbol{x}^0 \in \mathbb{R}^n$ be an initial solution and $\boldsymbol{x}^* \in \mathbb{R}^n$ be an arbitrary solution, where $R := \|\boldsymbol{x}^0 - \boldsymbol{x}^*\|_2$ and $R \ge \sqrt{n}$.[3]*

*Then, gradient descent with step size $\eta_t = 0.5 \min\left\{\frac{1}{\beta m^{-1}f(\boldsymbol{x})}, \frac{1}{\sqrt{\beta}\|\nabla f(\boldsymbol{x})\|_1}\right\}$ returns a solution with*

$$f(\boldsymbol{x}) \le (1+\delta)f(\boldsymbol{x}^*) + \varepsilon$$

*after*

$$T = O\left(\frac{\beta R^2}{m}\left(\frac{1}{\delta} + \log \frac{f(\boldsymbol{x}^0) - f(\boldsymbol{x}^*)}{\varepsilon}\right)\right)$$

*iterations.*

## 6 MAXIMUM MARGIN SOLUTIONS

It is known that running gradient descent on the logistic loss on linearly separable data converges to the hard SVM (maximum margin) classifier Soudry et al. (2018), yet at the slow rate of

$$\left\| \frac{\boldsymbol{x}^t}{\|\boldsymbol{x}^t\|_2} - \frac{\boldsymbol{x}^*}{\|\boldsymbol{x}^*\|_2} \right\|_2 \le O\left(\frac{\log \log t}{\log t}\right).$$

---

[3]The assumption $R \ge \sqrt{n}$ is not necessary but simplifies the bounds.

As all our results work best when the data is separable, it is natural to ask about what they imply for margin maximization.

We consider the constrained logistic regression problem

$$\min_{\|\boldsymbol{x}\|_2 \leq 1} f_p(\boldsymbol{x}) := \sum_i \log(1 + e^{-pb_i(\boldsymbol{Ax})_i}) \tag{4}$$

We start by observing that Corollary 5.1 and Theorem 5.2 can be modified to solve (4), with a blowup of $p^2$ in the number of iterations. In particular, the number of iterations will be $\widetilde{O}\left(p^2 X\left(\frac{1}{\delta} + \log\frac{1}{\varepsilon}\right)\right)$, where $X$ depends on whether we use Corollary 5.1 or Theorem 5.2, but is beyond the point of this section, since here we are interested in the error dependence.

Picking $\delta = 1$, $\varepsilon = me^{-p\alpha}$, and $p = \frac{\log(6m)}{\alpha\widehat{\varepsilon}}$ for some target error $\widehat{\varepsilon} \in (0, 1)$, we get the following theorem:

**Theorem 6.1.** *Consider a linearly separable binary classification instance $(\boldsymbol{A} \in \mathbb{R}^{m \times n}, \boldsymbol{b} \in \{1, -1\}^m)$, and a solution $\boldsymbol{x}^*$ that maximizes $\min_i \frac{b_i(\boldsymbol{Ax}^*)_i}{\|\boldsymbol{x}^*\|_2} := \alpha$. Then, we can obtain a solution $\boldsymbol{x}$ with $\|\boldsymbol{x}\|_2 \leq 1$ and $f_p(\boldsymbol{x}) \leq 3f_p(\boldsymbol{x}^*)$ in $\widetilde{O}\left(X\frac{1}{\alpha^2\widehat{\varepsilon}^3}\right)$ iterations of gradient descent, where $p = \frac{\log(6m)}{\alpha\widehat{\varepsilon}}$. Furthermore, $\boldsymbol{x}$ has $(1 - \widehat{\varepsilon})$-optimal margins:*

$$\min_i \frac{b_i(\boldsymbol{Ax})_i}{\|\boldsymbol{x}\|_2} \geq \alpha(1 - \widehat{\varepsilon})$$

*and is close to the maximum margin classifier:*

$$\left\|\frac{\boldsymbol{x}}{\|\boldsymbol{x}\|_2} - \frac{\boldsymbol{x}^*}{\|\boldsymbol{x}^*\|_2}\right\|_2 \leq 2\sqrt{\widehat{\varepsilon}}$$

It is not hard to see that Theorem 6.1 gives an exponential improvement in the error dependence compared to Soudry et al. (2018).

## 7   NUMERICAL EXAMPLE

In order to numerically validate our algorithm, we run logistic regression on the well known UCI adult binary classification dataset. In order to simulate a separable dataset, we first run gradient descent on the whole data, and then discard the misclassified data points. This gives us a separable dataset. Then, we run two variants of gradient descent: One with constant step size given by $\beta^{-1}$, and one with increasing step size given by $\eta_t = \beta^{-1}f(\boldsymbol{x}^0)/f(\boldsymbol{x}^t)$, with no other change. This is motivated by our findings, which suggest that the step size should increase proportionally to the decrease of the loss. As we can see in Figure 2, the error in the case of fixed step size decays as $\text{poly}(1/t)$, while in the case of increasing step size we have linear convergence (albeit with a low rate because the margins are in the order of $10^{-6}$).

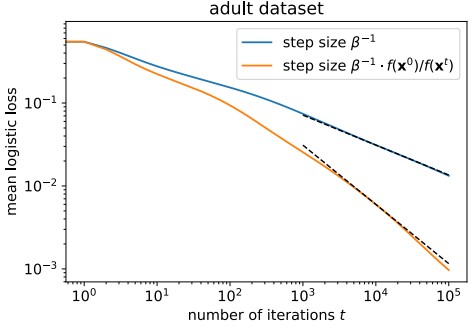

Figure 2: Comparison of fixed vs increasing step size on logistic regression on adult dataset

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

# A  MISSING PROOFS FROM SECTION 4

## A.1  PROOF OF MAIN LEMMA

**Lemma A.1** (Gradient lower bound). *Let $f : \mathbb{R}^n \to \mathbb{R}$ be a differentiable convex function and let $\boldsymbol{x} \in [-B', B']^n$, $\boldsymbol{x}^* \in [-B, B]^n$ be two solutions for some parameters $B' \geq B > 0$. For all $i \in [n]$ we define*

$$\zeta_i = \begin{cases} \lambda & \text{if } x_i = 0 \\ 0 & \text{if } |x_i| \geq B \text{ and } \nabla_i f(\boldsymbol{x}) \cdot x_i < 0 \\ 1 & \text{otherwise} \end{cases}$$

*where $0 < \lambda \leq 1$, and let $i^* = \operatorname{argmax}_i \{\zeta_i |\nabla_i f(\boldsymbol{x})|\}$. Then, at least one of the following is true:*

- $$|\nabla_{i^*} f(\boldsymbol{x})| \geq \frac{f(\boldsymbol{x}) - f(\boldsymbol{x}^*)}{\|\boldsymbol{x}^*\|_1 + \lambda \|\boldsymbol{x}\|_1}$$

- $$|\nabla_{i^*} f(\boldsymbol{x})| \geq \frac{f(\boldsymbol{x}) - f(\boldsymbol{x}^*)}{\lambda^{-1} \|\boldsymbol{x}^*\|_1 + \|\boldsymbol{x}\|_1} \text{ and } x_i \neq 0.$$

*Proof.* Let $S = \{i \mid x_i \neq 0\}$ and $F = \{i \mid |x_i| < B \text{ or } \nabla_i l(\boldsymbol{x}) \cdot x_i \geq 0\}$. By convexity of $f$, we have

$$\begin{aligned} f(\boldsymbol{x}^*) &\geq f(\boldsymbol{x}) + \langle \nabla f(\boldsymbol{x}), \boldsymbol{x}^* - \boldsymbol{x} \rangle \\ &\geq f(\boldsymbol{x}) + \langle \nabla_F f(\boldsymbol{x}), \boldsymbol{x}^* - \boldsymbol{x} \rangle \\ &= f(\boldsymbol{x}) + \langle \nabla_F f(\boldsymbol{x}), \boldsymbol{x}^* \rangle - \langle \nabla_{S \cap F} f(\boldsymbol{x}), \boldsymbol{x} \rangle \\ &\geq f(\boldsymbol{x}) - \|\nabla_F f(\boldsymbol{x})\|_\infty \|\boldsymbol{x}^*\|_1 - \|\nabla_{S \cap F} f(\boldsymbol{x})\|_\infty \|\boldsymbol{x}\|_1 \,, \end{aligned}$$

therefore

$$\|\nabla_F f(\boldsymbol{x})\|_\infty \|\boldsymbol{x}^*\|_1 + \|\nabla_{S \cap F} f(\boldsymbol{x})\|_\infty \|\boldsymbol{x}\|_1 \geq f(\boldsymbol{x}) - f(\boldsymbol{x}^*) \,. \tag{5}$$

Now, if $i^* \notin S$, by definition of the $\zeta_i$'s and $i^*$ we have

$$\lambda \|\nabla_F f(\boldsymbol{x})\|_\infty = \lambda |\nabla_{i^*} f(\boldsymbol{x})| \geq \|\nabla_{S \cap F} f(\boldsymbol{x})\|_\infty \,.$$

and so (5) implies

$$\begin{aligned} |\nabla_{i^*} f(\boldsymbol{x})| \|\boldsymbol{x}^*\|_1 + \lambda |\nabla_{i^*} f(\boldsymbol{x})| \|\boldsymbol{x}\|_1 &\geq f(\boldsymbol{x}) - f(\boldsymbol{x}^*) \\ \Rightarrow |\nabla_{i^*} f(\boldsymbol{x})| &\geq \frac{f(\boldsymbol{x}) - f(\boldsymbol{x}^*)}{\|\boldsymbol{x}^*\|_1 + \lambda \|\boldsymbol{x}\|_1} \,. \end{aligned}$$

Otherwise if $i^* \in S$, we have

$$\|\nabla_{S \cap F} f(\boldsymbol{x})\|_\infty = |\nabla_{i^*} f(\boldsymbol{x})| \geq \lambda \|\nabla_F f(\boldsymbol{x})\|_\infty \,,$$

and so (5) implies

$$\begin{aligned} \lambda^{-1} |\nabla_{i^*} f(\boldsymbol{x})| \|\boldsymbol{x}^*\|_1 + |\nabla_{i^*} f(\boldsymbol{x})| \|\boldsymbol{x}\|_1 &\geq f(\boldsymbol{x}) - f(\boldsymbol{x}^*) \\ \Rightarrow |\nabla_{i^*} f(\boldsymbol{x})| &\geq \frac{f(\boldsymbol{x}) - f(\boldsymbol{x}^*)}{\lambda^{-1} \|\boldsymbol{x}^*\|_1 + \|\boldsymbol{x}\|_1} \,. \end{aligned}$$

$\square$

**Lemma A.2** (Coordinate update). *Let $f : \mathbb{R}^n \to \mathbb{R}_{\geq 0}$ be a twice continuously differentiable convex function that is $2\gamma$-second order robust and $\gamma^2$-multiplicatively smooth with respect to the $\ell_1$ norm, for some $\gamma > 0$. Let $\boldsymbol{x} \in [-B', B']^n$ be a suboptimal solution such that $f(\boldsymbol{x}) \geq f(\boldsymbol{x}^*)$, where $\boldsymbol{x}^* \in [-B, B]^n$ is some unknown solution with $\gamma \|\boldsymbol{x}^*\|_1 \geq 1$, and $B' \geq B > 0$ are some parameters. We make the update*

$$\boldsymbol{x}' = \boldsymbol{x} - \eta \nabla_i f(\boldsymbol{x}) \mathbf{1}_i \,,$$

*where $i$ is picked as in Lemma A.1 for some parameter $\lambda \in (0,1)$ and $\eta = 0.5 \min \left\{ \frac{1}{\gamma^2 f(\boldsymbol{x})}, \frac{1}{\gamma |\nabla_i f(\boldsymbol{x})|} \right\}$ is a step size. Then, at least one of the following is true about the progress in decreasing $f$:*

- $f(\boldsymbol{x}) - f(\boldsymbol{x}') \geq \dfrac{(f(\boldsymbol{x}) - f(\boldsymbol{x}^*))^2}{4\gamma^2 f(\boldsymbol{x}) \left( \|\boldsymbol{x}^*\|_1 + \lambda \|\boldsymbol{x}\|_1 \right)^2}$

- $f(\boldsymbol{x}) - f(\boldsymbol{x}') \geq \dfrac{(f(\boldsymbol{x}) - f(\boldsymbol{x}^*))^2}{4\gamma^2 f(\boldsymbol{x}) \left( \lambda^{-1} \|\boldsymbol{x}^*\|_1 + \|\boldsymbol{x}\|_1 \right)^2}$ *and $x_i \neq 0$,*

*and the norm of the new solution is bounded as $\|\boldsymbol{x}'\|_\infty \leq \max \{ B', B + \frac{1}{2\gamma} \}$. In the case that $f(\boldsymbol{x}) < f(\boldsymbol{x}^*)$ we have $f(\boldsymbol{x}') \leq f(\boldsymbol{x})$.*

*Proof.* We first consider a generic update $\boldsymbol{x}' = \boldsymbol{x} + \widetilde{\boldsymbol{x}}$. By Taylor's theorem and the fact that $f$ is twice continuously differentiable, we have

$$f(\boldsymbol{x}') = f(\boldsymbol{x}) + \langle \nabla f(\boldsymbol{x}), \widetilde{\boldsymbol{x}} \rangle + \frac{1}{2} \langle \widetilde{\boldsymbol{x}}, \nabla^2 f(\bar{\boldsymbol{x}}) \widetilde{\boldsymbol{x}} \rangle \,,$$

for some $\bar{\boldsymbol{x}}$ that is entrywise between $\boldsymbol{x}$ and $\widetilde{\boldsymbol{x}}$.

Since $f$ is $2\gamma$-second-order-robust and $\gamma^2$-multiplicatively-smooth with respect to the $\ell_1$ norm, as long as the update is bounded in $\ell_1$ norm as

$$\|\widetilde{\boldsymbol{x}}\|_1 \leq 1/(2\gamma) \tag{6}$$

we have

$$f(\boldsymbol{x}') \leq f(\boldsymbol{x}) + \langle \nabla f(\boldsymbol{x}), \widetilde{\boldsymbol{x}} \rangle + \langle \widetilde{\boldsymbol{x}}, \nabla^2 f(\boldsymbol{x}) \widetilde{\boldsymbol{x}} \rangle$$
$$\leq f(\boldsymbol{x}) + \langle \nabla f(\boldsymbol{x}), \widetilde{\boldsymbol{x}} \rangle + \gamma^2 f(\boldsymbol{x}) \|\widetilde{\boldsymbol{x}}\|_1^2 \,.$$

Note that the right hand side is minimized for

$$\widetilde{\boldsymbol{x}} = -\frac{H_1 \left( \nabla f(\boldsymbol{x}) \right)}{2\gamma^2 f(\boldsymbol{x})} \,,$$

where $H_1$ is the hard thresholding operator that zeroes out all but the top entry in absolute value. This is a coordinate descent step. Our step will be slightly more careful so that it doesn't unnecessarily increase the sparsity of $\boldsymbol{x}$. We consider the following coordinate step

$$\widetilde{\boldsymbol{x}} = -\eta \nabla_i f(\boldsymbol{x}) \,,$$

where $\eta > 0$ and $i$ are as defined in the lemma statement. We now have

$$f(\boldsymbol{x}) - f(\boldsymbol{x}') \geq \left( \eta - \eta^2 \gamma^2 f(\boldsymbol{x}) \right) \left( \nabla_i f(\boldsymbol{x}) \right)^2$$

The term $\left( \eta - \eta^2 \gamma^2 f(\boldsymbol{x}) \right)$ is maximized at $\eta = \frac{1}{2\gamma^2 f(\boldsymbol{x})}$. In addition, to stay in the $\ell_1$ neighborhood where the Hessian in stable, we need to satisfy (6) by making sure that $\eta \leq \frac{1}{2\gamma |\nabla_i f(\boldsymbol{x})|}$. Based on these requirements, we pick

$$\eta = \min \left\{ \frac{1}{2\gamma^2 f(\boldsymbol{x})}, \frac{1}{2\gamma |\nabla_i f(\boldsymbol{x})|} \right\}$$

and conclude that

$$f(\boldsymbol{x}) - f(\boldsymbol{x}') \geq \min \left\{ \frac{1}{4\gamma^2 f(\boldsymbol{x})}, \frac{1}{4\gamma |\nabla_i f(\boldsymbol{x})|} \right\} \left( \nabla_i f(\boldsymbol{x}) \right)^2$$
$$= \min \left\{ \frac{(\nabla_i f(\boldsymbol{x}))^2}{4\gamma^2 f(\boldsymbol{x})}, \frac{|\nabla_i f(\boldsymbol{x})|}{4\gamma} \right\} \,.$$

Note that this is always $\geq 0$ and so we have $f(\boldsymbol{x}') \leq f(\boldsymbol{x})$ even if $f(\boldsymbol{x}) < f(\boldsymbol{x}^*)$. We now take two cases and use the two bullets of Lemma A.1 accordingly.

**Case 1:** $x_i = 0$. The first bullet of Lemma A.1 has to be true, i.e.

$$|\nabla_i f(\boldsymbol{x})| \geq \frac{f(\boldsymbol{x}) - f(\boldsymbol{x}^*)}{\|\boldsymbol{x}^*\|_1 + \lambda \|\boldsymbol{x}\|_1} \, .$$

Therefore,

$$f(\boldsymbol{x}) - f(\boldsymbol{x}') \geq \min \left\{ \frac{(f(\boldsymbol{x}) - f(\boldsymbol{x}^*))^2}{4\gamma^2 f(\boldsymbol{x}) \left(\|\boldsymbol{x}^*\|_1 + \lambda \|\boldsymbol{x}\|_1\right)^2}, \frac{f(\boldsymbol{x}) - f(\boldsymbol{x}^*)}{4\gamma \left(\|\boldsymbol{x}^*\|_1 + \lambda \|\boldsymbol{x}\|_1\right)} \right\}$$

$$= \frac{(f(\boldsymbol{x}) - f(\boldsymbol{x}^*))^2}{4\gamma^2 f(\boldsymbol{x}) \left(\|\boldsymbol{x}^*\|_1 + \lambda \|\boldsymbol{x}\|_1\right)^2} \, ,$$

where we used the facts that $f(\boldsymbol{x}) - f(\boldsymbol{x}^*) \leq f(\boldsymbol{x})$ and $\gamma \|\boldsymbol{x}^*\|_1 \geq 1$.

**Case 2:** $x_i \neq 0$. If the first bullet of Lemma A.1 is true, we can proceed as in the previous case. Otherwise, we use the second bullet of Lemma A.1 and similarly get

$$f(\boldsymbol{x}) - f(\boldsymbol{x}') \geq \frac{(f(\boldsymbol{x}) - f(\boldsymbol{x}^*))^2}{4\gamma^2 f(\boldsymbol{x}) \left(\lambda^{-1} \|\boldsymbol{x}^*\|_1 + \|\boldsymbol{x}\|_1\right)^2} \, .$$

Finally, in order to bound $\|\boldsymbol{x}'\|_\infty$, we first note that $\|\boldsymbol{x}\|_\infty \leq B'$. Now, by our choice of $i$ we have that either $|x_i| < B$, or $\nabla_i f(\boldsymbol{x}) \cdot x_i > 0$. In the first case, we have

$$|x_i'| \leq |x_i| + |\widetilde{x}_i| < B + \frac{1}{2\gamma} \, ,$$

where we used (6). Otherwise, we have that $|x_i| \geq B$ and $\nabla_i f(\boldsymbol{x}) \cdot x_i > 0$. This implies that $x_i$ and $\widetilde{x}_i$ have different signs, so

$$|x_i'| = |x_i + \widetilde{x}_i| \leq \max\left\{|x_i|, |\widetilde{x}_i|\right\} \leq \max\left\{B', \frac{1}{2\gamma}\right\} \, .$$

Therefore, in any case we have $|x_i'| \leq \max\left\{B', B + \frac{1}{2\gamma}\right\}$. $\qquad\square$

## A.2 Proofs of theorems

### A.2.1 Proof of Corollary 5.1

*Proof.* We will apply Lemma A.2 for $T$ iterations to obtain solutions $\boldsymbol{x}^0, \ldots, \boldsymbol{x}^T$, where for some $T$ that will be defined later. The logistic function $f$ is $2M$-second order robust and $M^2$-multiplicatively smooth with respect to the $\ell_1$ norm, so Lemma A.2 can be applied with $\gamma = M$ and $B' = B + \frac{1}{2M}$.

Based on the guarantee of Lemma A.2, we get the following bound on the $\ell_1$ norm of $\boldsymbol{x}^t$ at all times:

$$\|\boldsymbol{x}^t\|_1 \leq n \|\boldsymbol{x}^t\|_\infty \leq n \left(B + \frac{1}{2M}\right) \leq (3/2)nB \, .$$

Let $\bar{t}$ be the smallest $t \geq 0$ for which $f(\boldsymbol{x}^{\bar{t}}) \leq 2f(\boldsymbol{x}^*)$ or $f(\boldsymbol{x}^{\bar{t}}) \leq f(\boldsymbol{x}^*) + \varepsilon$, and let $\bar{t} = \infty$ if this never happens. Therefore, for all $t < \bar{t}$ we have $f(\boldsymbol{x}^t) \geq 2f(\boldsymbol{x}^*) \Rightarrow \frac{f(\boldsymbol{x}^t) - f(\boldsymbol{x}^*)}{f(\boldsymbol{x}^t)} \geq \frac{1}{2}$, and so the statement of Lemma A.2 gives:

$$f(\boldsymbol{x}^t) - f(\boldsymbol{x}^{t+1}) \geq \frac{f(\boldsymbol{x}^t) - f(\boldsymbol{x}^*)}{8M^2(\|\boldsymbol{x}^*\|_1 + \|\boldsymbol{x}^t\|_1)^2}$$

$$\geq \frac{f(\boldsymbol{x}^t) - f(\boldsymbol{x}^*)}{8n^2 M^2 (B + (3/2)B)^2}$$

$$\geq \frac{f(\boldsymbol{x}^t) - f(\boldsymbol{x}^*)}{50 n^2 M^2 B^2} \, ,$$

where we used the fact that $\|\boldsymbol{x}^*\|_1 \leq n \|\boldsymbol{x}^*\|_\infty \leq nB$. Equivalently,

$$f(\boldsymbol{x}^{t+1}) - f(\boldsymbol{x}^*) \leq \left(1 - \frac{1}{50 n^2 M^2 B^2}\right) (f(\boldsymbol{x}^t) - f(\boldsymbol{x}^*)) \, ,$$

and summing up these for $t \in \{0, 1, \ldots, \bar{t} - 1\}$, we get

$$f(\boldsymbol{x}^{\bar{t}}) - f(\boldsymbol{x}^*) \leq \left(1 - \frac{1}{50n^2 M^2 B^2}\right)^{\bar{t}} (f(\boldsymbol{x}^0) - f(\boldsymbol{x}^*))$$

$$\leq \varepsilon,$$

as long as

$$\bar{t} \geq 50n^2 M^2 B^2 \log \frac{f(\boldsymbol{x}^0) - f(\boldsymbol{x}^*)}{\varepsilon},$$

therefore we conclude that $\bar{t}$ is at most this quantity.

Now we consider the iterations $t \geq \bar{t}$. If $f(\boldsymbol{x}^{\bar{t}}) \leq f(\boldsymbol{x}^*) + \varepsilon$ there are no such iterations and we are done. Therefore we have that $f(\boldsymbol{x}^{\bar{t}}) \leq 2f(\boldsymbol{x}^*)$. We again use Lemma A.2 for all such $t$, which gives

$$f(\boldsymbol{x}^t) - f(\boldsymbol{x}^{t+1}) \geq \frac{(f(\boldsymbol{x}^t) - f(\boldsymbol{x}^*))^2}{4M^2 f(\boldsymbol{x}^t)(\|\boldsymbol{x}^*\|_1 + \|\boldsymbol{x}^t\|_1)^2}$$

$$\geq \frac{(f(\boldsymbol{x}^t) - f(\boldsymbol{x}^*))^2}{25f(\boldsymbol{x}^t)n^2 M^2 B^2}$$

$$\geq \frac{(f(\boldsymbol{x}^t) - f(\boldsymbol{x}^*))^2}{50f(\boldsymbol{x}^*)n^2 M^2 B^2}.$$

By known convergence results, this recurrence leads to the bound

$$f(\boldsymbol{x}^T) \leq f(\boldsymbol{x}^*) + \frac{100f(\boldsymbol{x}^*)n^2 M^2 B^2}{T - \bar{t}}$$

$$\leq f(\boldsymbol{x}^*) \left(1 + \frac{100n^2 M^2 B^2}{T - \bar{t}}\right),$$

implying that $f(\boldsymbol{x}^T) \leq (1 + \delta)f(\boldsymbol{x}^*)$ after

$$T - \bar{t} = O\left(n^2 M^2 B^2 \frac{1}{\delta}\right)$$

additional iterations after $\bar{t}$. Therefore, the total number of iterations to achieve $f(\boldsymbol{x}^T) \leq (1 + \delta) \cdot f(\boldsymbol{x}^*) + \varepsilon$ is

$$O\left(n^2 M^2 B^2 \left(\frac{1}{\delta} + \log \frac{f(\boldsymbol{x}^0) - f(\boldsymbol{x}^*)}{\varepsilon}\right)\right).$$

$\square$

### A.2.2 PROOF OF THEOREM 4.1

*Proof.* Similarly to the proof of Corollary 5.1, we apply Lemma A.2 for $T$ iterations to obtain solutions $\boldsymbol{x}^0, \ldots, \boldsymbol{x}^T$, but now we also have to account for the sparsity increase of $\boldsymbol{x}^t$. For this reason, we use $\lambda_t < 1$, which disincentivizes updating zero entries of the solution vector.

Compared to Corollary 5.1, we have the differences that

$$\lambda_t^{-1} \|\boldsymbol{x}^*\|_1 = \max\left\{c^{-1} \|\boldsymbol{x}^t\|_1, \|\boldsymbol{x}^*\|_1\right\},$$

and that we have the following tighter bounds because of sparsity:

$$\|\boldsymbol{x}^*\|_1 \leq sB$$

$$\|\boldsymbol{x}^t\|_1 \leq \|\boldsymbol{x}^t\|_0 \|\boldsymbol{x}^t\|_\infty \leq \|\boldsymbol{x}^t\|_0 (3/2)B.$$

We first bound the sparsity. Note that the sparsity increases by at most 1 every time the first bullet of Lemma A.2 is true, and does not increase when the second bullet is true. Therefore, the progress in each sparsity-increasing iteration is

$$f(\boldsymbol{x}^t) - f(\boldsymbol{x}^{t+1}) \geq \frac{(f(\boldsymbol{x}^t) - f(\boldsymbol{x}^*))^2}{4f(\boldsymbol{x}^*)M^2 (\|\boldsymbol{x}^*\|_1 + \lambda_t \|\boldsymbol{x}^t\|_1)^2}$$

$$\geq \frac{(f(\boldsymbol{x}^t) - f(\boldsymbol{x}^*))^2}{4(1 + c)^2 f(\boldsymbol{x}^*)M^2 \|\boldsymbol{x}^*\|_1^2}.$$

Completely analogously to the proof of Corollary 5.1, this implies that the total number of such iterations (and thus total sparsity) is

$$s' := \left\| \boldsymbol{x}^T \right\|_0 \leq O\left( \left\| \boldsymbol{x}^* \right\|_1^2 (1+c)^2 M^2 \left( \frac{1}{\delta} + \log \frac{f(\boldsymbol{x}^0) - f(\boldsymbol{x}^*)}{\varepsilon} \right) \right) .$$

Now it remains to bound the total number of iterations. We have

$$\max \left\{ \left\| \boldsymbol{x}^* \right\|_1 + \lambda_t \left\| \boldsymbol{x}^t \right\|_1, \lambda_t^{-1} \left\| \boldsymbol{x}^* \right\|_1 + \left\| \boldsymbol{x}^t \right\|_1 \right\}$$
$$\leq \max \left\{ \left\| \boldsymbol{x}^* \right\|_1 + \left\| \boldsymbol{x}^t \right\|_1, c^{-1} \left\| \boldsymbol{x}^t \right\|_1 + \left\| \boldsymbol{x}^t \right\|_1 \right\}$$
$$\leq \left\| \boldsymbol{x}^* \right\|_1 + (1 + c^{-1}) \left\| \boldsymbol{x}^t \right\|_1$$
$$\leq \left\| \boldsymbol{x}^* \right\|_1 + \frac{3}{2}(1 + c^{-1}) \left\| \boldsymbol{x}^t \right\|_0 B$$
$$\leq \left\| \boldsymbol{x}^* \right\|_1 + \frac{3}{2}(1 + c^{-1}) \left\| \boldsymbol{x}^T \right\|_0 B .$$

As a result, the progress bound of Lemma A.2 becomes

$$f(\boldsymbol{x}^t) - f(\boldsymbol{x}^{t+1}) \geq \frac{(f(\boldsymbol{x}) - f(\boldsymbol{x}^*))^2}{4f(\boldsymbol{x})M^2 \left( \left\| \boldsymbol{x}^* \right\|_1 + \frac{3}{2}(1 + c^{-1}) \left\| \boldsymbol{x}^T \right\|_0 B \right)^2} ,$$

and, analogously to the proof of Corollary 5.1 and using the fact that $\left\| \boldsymbol{x}^* \right\|_1 \leq \left\| \boldsymbol{x}^* \right\|_0 B$, the total number of iterations is bounded by

$$T = O\left( (1 + c^{-1})^2 \left( \left\| \boldsymbol{x}^T \right\|_0^2 + \left\| \boldsymbol{x}^* \right\|_0^2 \right) M^2 B^2 \left( \frac{1}{\delta} + \log \frac{f(\boldsymbol{x}^0) - f(\boldsymbol{x}^*)}{\varepsilon} \right) \right) .$$

$\square$

### A.2.3 PROOF OF THEOREM 4.3

*Proof.* We move similarly to the proof of Theorem 4.1, but now we can strengthen Lemma A.2 because $\boldsymbol{x}^t$ is fully corrected for all $t$, i.e. $\nabla_i f(\boldsymbol{x}^t) = 0$ for all $i \in \operatorname{supp}(\boldsymbol{x}^t)$. As in the proof of Lemma A.2, we can lower bound the amount of progress as a function of $\left\| \nabla f(\boldsymbol{x}^t) \right\|_\infty$ as follows:

$$f(\boldsymbol{x}^t) - f(\boldsymbol{x}^{t+1}) \geq \min \left\{ \frac{(\nabla_i f(\boldsymbol{x}^t))^2}{4M^2 f(\boldsymbol{x}^t)}, \frac{|\nabla_i f(\boldsymbol{x}^t)|}{4M} \right\} .$$

Now, by convexity of $f$ we have

$$\langle \nabla f(\boldsymbol{x}^t), \boldsymbol{x}^t - \boldsymbol{x}^* \rangle \geq f(\boldsymbol{x}^t) - f(\boldsymbol{x}^*) . \tag{7}$$

Because of fully corrective steps we have $\langle \nabla f(\boldsymbol{x}^t), \boldsymbol{x}^t \rangle = 0$, and so the left hand side of (7) is upper bounded by $\left\| \nabla f(\boldsymbol{x}^t) \right\|_\infty \left\| \boldsymbol{x}^* \right\|_1$. As a result, we have

$$\left\| \nabla f(\boldsymbol{x}^t) \right\|_\infty^2 \geq \frac{(f(\boldsymbol{x}) - f(\boldsymbol{x}^*))^2}{\left\| \boldsymbol{x}^* \right\|_1^2} ,$$

and so we get the progress bound of

$$f(\boldsymbol{x}^t) - f(\boldsymbol{x}^{t+1}) \geq \min \left\{ \frac{(f(\boldsymbol{x}^t) - f(\boldsymbol{x}^*))^2}{4M^2 f(\boldsymbol{x}^t) \left\| \boldsymbol{x}^* \right\|_1^2}, \frac{f(\boldsymbol{x}^t) - f(\boldsymbol{x}^*)}{4M \left\| \boldsymbol{x}^* \right\|_1} \right\}$$
$$\geq \frac{(f(\boldsymbol{x}^t) - f(\boldsymbol{x}^*))^2}{4M^2 f(\boldsymbol{x}^t) \left\| \boldsymbol{x}^* \right\|_1^2} ,$$

because $M \left\| \boldsymbol{x}^* \right\|_1 > 1$. Similarly to the proof of Theorem 4.1, this progress bound leads to a sparsity of

$$s' := \left\| \boldsymbol{x}^T \right\|_0 \leq O\left( \left\| \boldsymbol{x}^* \right\|_1^2 M^2 \left( \frac{1}{\delta} + \log \frac{f(\boldsymbol{x}^0) - f(\boldsymbol{x}^*)}{\varepsilon} \right) \right)$$

and the same number of iterations. $\square$

# B MISSING PROOFS FROM SECTION 5

## B.1 GRADIENT UPDATE LEMMA

**Lemma B.1** (Gradient update). *Let $f : \mathbb{R}^n \to \mathbb{R}_{>0}$ be a twice continuously differentiable convex function that is $2\gamma$-second order robust with respect to the $\ell_1$ norm and $\mu$-multiplicatively smooth with respect to the $\ell_2$ norm for some $\gamma, \mu > 0$. Let $\boldsymbol{x} \in \mathbb{R}^n$ be a solution such that $f(\boldsymbol{x}) > f(\boldsymbol{x}^*)$, where $\boldsymbol{x}^* \in \mathbb{R}^n$ is an unknown solution with $\|\boldsymbol{x} - \boldsymbol{x}^*\|_2 \leq \|\boldsymbol{x}^*\|_2$. We make the update*

$$\boldsymbol{x}' = \boldsymbol{x} - \eta \nabla f(\boldsymbol{x}),$$

*where $\eta = 0.5 \min \left\{ \frac{1}{\mu f(\boldsymbol{x})}, \frac{1}{\gamma \|\nabla f(\boldsymbol{x})\|_1} \right\}$ is a step size. Then, the progress in decreasing $f$ is:*

$$f(\boldsymbol{x}) - f(\boldsymbol{x}') \geq \min \left\{ \frac{(f(\boldsymbol{x}) - f(\boldsymbol{x}^*))^2}{4\mu f(\boldsymbol{x}) \|\boldsymbol{x}^*\|_2^2}, \frac{f(\boldsymbol{x}) - f(\boldsymbol{x}^*)}{4\gamma\sqrt{n} \|\boldsymbol{x}^*\|_2} \right\}.$$

*Additionally, as long as $\boldsymbol{x}'$ is still suboptimal with respect to $\boldsymbol{x}^*$, i.e. $f(\boldsymbol{x}') > f(\boldsymbol{x}^*)$, the distance to $\boldsymbol{x}^*$ decreases: $\|\boldsymbol{x}' - \boldsymbol{x}^*\|_2 \leq \|\boldsymbol{x} - \boldsymbol{x}^*\|_2$. Finally, if $f(\boldsymbol{x}) \leq f(\boldsymbol{x}^*)$, then $f(\boldsymbol{x}') \leq f(\boldsymbol{x})$.*

*Proof.* We first consider a generic update $\boldsymbol{x}' = \boldsymbol{x} + \widetilde{\boldsymbol{x}}$. By Taylor's theorem and the fact that $f$ is twice continuously differentiable, we have

$$f(\boldsymbol{x}') = f(\boldsymbol{x}) + \langle \nabla f(\boldsymbol{x}), \widetilde{\boldsymbol{x}} \rangle + \frac{1}{2} \langle \widetilde{\boldsymbol{x}}, \nabla^2 f(\bar{\boldsymbol{x}}) \widetilde{\boldsymbol{x}} \rangle,$$

for some $\bar{\boldsymbol{x}}$ that is entrywise between $\boldsymbol{x}$ and $\boldsymbol{x}'$.

Since $f$ is $2\gamma$-second-order-robust with respect to $\ell_1$ and and $\mu$-multiplicatively-smooth with respect to the $\ell_2$ norm, as long as the update is bounded in $\ell_1$ norm as

$$\|\widetilde{\boldsymbol{x}}\|_1 \leq 1/(2\gamma) \tag{8}$$

we have

$$f(\boldsymbol{x}') \leq f(\boldsymbol{x}) + \langle \nabla f(\boldsymbol{x}), \widetilde{\boldsymbol{x}} \rangle + \langle \widetilde{\boldsymbol{x}}, \nabla^2 f(\boldsymbol{x}) \widetilde{\boldsymbol{x}} \rangle$$
$$\leq f(\boldsymbol{x}) + \langle \nabla f(\boldsymbol{x}), \widetilde{\boldsymbol{x}} \rangle + \mu f(\boldsymbol{x}) \|\widetilde{\boldsymbol{x}}\|_2^2.$$

Note that the right hand side is minimized for

$$\widetilde{\boldsymbol{x}} = -\frac{1}{2\mu f(\boldsymbol{x})} \nabla f(\boldsymbol{x}).$$

In addition, to stay in the $\ell_1$ neighborhood where the Hessian in stable, we need to satisfy (8). Based on these requirements, we make the update $\widetilde{\boldsymbol{x}} = -\eta \nabla f(\boldsymbol{x})$, where

$$\eta = \min \left\{ \frac{1}{2\mu f(\boldsymbol{x})}, \frac{1}{2\gamma \|\nabla f(\boldsymbol{x})\|_1} \right\}.$$

We thus have

$$f(\boldsymbol{x}) - f(\boldsymbol{x}') \geq \left( \eta - \eta^2 \mu f(\boldsymbol{x}) \right) \|\nabla f(\boldsymbol{x})\|_2^2$$
$$\geq \frac{\eta}{2} \|\nabla f(\boldsymbol{x})\|_2^2$$

and so

$$f(\boldsymbol{x}) - f(\boldsymbol{x}') \geq \min \left\{ \frac{1}{4\mu f(\boldsymbol{x})}, \frac{1}{4\gamma \|\nabla f(\boldsymbol{x})\|_1} \right\} \|\nabla f(\boldsymbol{x})\|_2^2$$
$$\geq \min \left\{ \frac{\|\nabla f(\boldsymbol{x})\|_2^2}{4\mu f(\boldsymbol{x})}, \frac{\|\nabla f(\boldsymbol{x})\|_2}{4\gamma\sqrt{n}} \right\}.$$

This takes care of the case $f(\boldsymbol{x}) \leq f(\boldsymbol{x}^*)$, since it shows that $f(\boldsymbol{x}') \leq f(\boldsymbol{x})$. Now we deal with the case $f(\boldsymbol{x}) > f(\boldsymbol{x}^*)$. By convexity we have

$$
\begin{aligned}
f(\boldsymbol{x}^*) &\geq f(\boldsymbol{x}) + \langle \nabla f(\boldsymbol{x}), \boldsymbol{x}^* - \boldsymbol{x} \rangle \\
&\geq f(\boldsymbol{x}) - \|\nabla f(\boldsymbol{x})\|_2 \|\boldsymbol{x}^* - \boldsymbol{x}\|_2 \\
&\geq f(\boldsymbol{x}) - \|\nabla f(\boldsymbol{x})\|_2 \|\boldsymbol{x}^*\|_2 \,,
\end{aligned}
$$

which gives

$$
\|\nabla f(\boldsymbol{x})\|_2^2 \geq \frac{(f(\boldsymbol{x}) - f(\boldsymbol{x}^*))^2}{\|\boldsymbol{x}^*\|_2^2} \,,
$$

and so

$$
f(\boldsymbol{x}) - f(\boldsymbol{x}') \geq \min \left\{ \frac{(f(\boldsymbol{x}) - f(\boldsymbol{x}^*))^2}{4\mu f(\boldsymbol{x}) \|\boldsymbol{x}^*\|_2^2}, \frac{f(\boldsymbol{x}) - f(\boldsymbol{x}^*)}{4\gamma\sqrt{n} \|\boldsymbol{x}^*\|_2} \right\} \,.
$$

For the norm bound, we suppose that $f(\boldsymbol{x}') > f(\boldsymbol{x}^*)$ (otherwise we are done). We have

$$
\begin{aligned}
&\|\boldsymbol{x}' - \boldsymbol{x}^*\|_2^2 - \|\boldsymbol{x} - \boldsymbol{x}^*\|_2^2 \\
&= \|\boldsymbol{x}' - \boldsymbol{x}\|_2^2 + 2\langle \boldsymbol{x} - \boldsymbol{x}^*, \boldsymbol{x}' - \boldsymbol{x} \rangle \\
&= \eta^2 \|\nabla f(\boldsymbol{x})\|_2^2 - 2\eta \langle \boldsymbol{x} - \boldsymbol{x}^*, \nabla f(\boldsymbol{x}) \rangle \,.
\end{aligned}
$$

Now, note that

$$
\frac{\eta}{2} \|\nabla f(\boldsymbol{x})\|_2^2 \leq f(\boldsymbol{x}) - f(\boldsymbol{x}') \leq f(\boldsymbol{x}) - f(\boldsymbol{x}^*)
$$

and by convexity $\langle \boldsymbol{x} - \boldsymbol{x}^*, \nabla f(\boldsymbol{x}) \rangle \geq f(\boldsymbol{x}) - f(\boldsymbol{x}^*)$, so

$$
\begin{aligned}
&\|\boldsymbol{x}' - \boldsymbol{x}^*\|_2^2 - \|\boldsymbol{x} - \boldsymbol{x}^*\|_2^2 \\
&= \eta^2 \|\nabla f(\boldsymbol{x})\|_2^2 - 2\eta \langle \boldsymbol{x} - \boldsymbol{x}^*, \nabla f(\boldsymbol{x}) \rangle \\
&\leq 0 \,.
\end{aligned}
$$

$\square$

## B.2 Proof of Theorem 5.2

*Proof.* We repeatedly use Lemma B.1 to obtain iterates $\boldsymbol{x}^0, \boldsymbol{x}^1, \ldots, \boldsymbol{x}^T$. Note that as long as $f(\boldsymbol{x}^t) > f(\boldsymbol{x}^*)$, we have $\|\boldsymbol{x}^t - \boldsymbol{x}^*\|_2 \leq \|\boldsymbol{x}^0 - \boldsymbol{x}^*\|_2 := R$.

Let $\bar{t}$ be the smallest $t \geq 0$ for which $f(\boldsymbol{x}^{\bar{t}}) \leq 2f(\boldsymbol{x}^*)$ or $f(\boldsymbol{x}^{\bar{t}}) \leq f(\boldsymbol{x}^*) + \varepsilon$, and let $\bar{t} = \infty$ if this never happens. Therefore, for all $t < \bar{t}$ we have $f(\boldsymbol{x}^t) \geq 2f(\boldsymbol{x}^*) \Rightarrow \frac{f(\boldsymbol{x}^t) - f(\boldsymbol{x}^*)}{f(\boldsymbol{x}^t)} \geq \frac{1}{2}$, and so the statement of Lemma B.1 gives:

$$
\begin{aligned}
f(\boldsymbol{x}^t) - f(\boldsymbol{x}^{t+1}) &\geq \min \left\{ \frac{1}{8\mu \|\boldsymbol{x}^*\|_2^2}, \frac{1}{4\gamma\sqrt{n} \|\boldsymbol{x}^*\|_2} \right\} \cdot (f(\boldsymbol{x}^t) - f(\boldsymbol{x}^*)) \\
&\geq \frac{1}{8\mu \|\boldsymbol{x}^*\|_2^2 + 4\gamma\sqrt{n} \|\boldsymbol{x}^*\|_2} \cdot (f(\boldsymbol{x}^t) - f(\boldsymbol{x}^*)) \,.
\end{aligned}
$$

Equivalently,

$$
f(\boldsymbol{x}^{t+1}) - f(\boldsymbol{x}^*) \leq \left( 1 - \frac{1}{8\mu \|\boldsymbol{x}^*\|_2^2 + 4\gamma\sqrt{n} \|\boldsymbol{x}^*\|_2} \right) (f(\boldsymbol{x}^t) - f(\boldsymbol{x}^*)) \,,
$$

and summing up these for $t \in \{0, 1, \ldots, \bar{t} - 1\}$, we get

$$
\begin{aligned}
f(\boldsymbol{x}^{\bar{t}}) - f(\boldsymbol{x}^*) &\leq \left( 1 - \frac{1}{8\mu \|\boldsymbol{x}^*\|_2^2 + 4\gamma\sqrt{n} \|\boldsymbol{x}^*\|_2} \right)^{\bar{t}} (f(\boldsymbol{x}^0) - f(\boldsymbol{x}^*)) \\
&\leq \varepsilon \,,
\end{aligned}
$$

as long as

$$\bar{t} \geq \left(8\mu R^2 + 4\gamma\sqrt{n}R\right) \log \frac{f(\boldsymbol{x}^0) - f(\boldsymbol{x}^*)}{\varepsilon},$$

therefore we conclude that $\bar{t}$ is at most this quantity.

Now we consider the iterations $t \geq \bar{t}$. If $f(\boldsymbol{x}^{\bar{t}}) \leq f(\boldsymbol{x}^*) + \varepsilon$ there are no such iterations and we are done. Therefore we have that $f(\boldsymbol{x}^{\bar{t}}) \leq 2f(\boldsymbol{x}^*)$. We again use Lemma B.1 for all such $t$, which gives

$$f(\boldsymbol{x}^t) - f(\boldsymbol{x}^{t+1}) \geq \frac{(f(\boldsymbol{x}^t) - f(\boldsymbol{x}^*))^2}{4\mu f(\boldsymbol{x}^t)R^2}$$

$$\geq \frac{(f(\boldsymbol{x}^t) - f(\boldsymbol{x}^*))^2}{8\mu f(\boldsymbol{x}^*)R^2}.$$

By known convergence results, this recurrence leads to the bound

$$f(\boldsymbol{x}^T) \leq f(\boldsymbol{x}^*) + \frac{16\mu f(\boldsymbol{x}^*)R^2}{T - \bar{t}}$$

$$= f(\boldsymbol{x}^*)\left(1 + \frac{16\mu R^2}{T - \bar{t}}\right),$$

implying that $f(\boldsymbol{x}^T) \leq (1 + \delta)f(\boldsymbol{x}^*)$ after

$$T - \bar{t} = O\left(\mu R^2 \frac{1}{\delta}\right)$$

additional iterations after $\bar{t}$.

Therefore, the total number of iterations to achieve $f(\boldsymbol{x}^T) \leq (1 + \delta) \cdot f(\boldsymbol{x}^*) + \varepsilon$ is

$$O\left(\left(\mu R^2 + \gamma\sqrt{n}R\right)\left(\frac{1}{\delta} + \log \frac{f(\boldsymbol{x}^0) - f(\boldsymbol{x}^*)}{\varepsilon}\right)\right).$$

For $\mu = \beta m^{-1}$, $\gamma = \sqrt{\beta}$, and using the fact that $R \geq \sqrt{n}$, we get

$$O\left(\frac{\beta R^2}{m}\left(\frac{1}{\delta} + \log \frac{f(\boldsymbol{x}^0) - f(\boldsymbol{x}^*)}{\varepsilon}\right)\right)$$

iterations. □

## C  PROOF OF THEOREM 6.1

*Proof.* Let us consider a classifier $\boldsymbol{x}^*$ with $\|\boldsymbol{x}^*\|_2 = 1$ and margins $\geq \alpha$, i.e. $b_i(\boldsymbol{A}\boldsymbol{x}^*)_i \geq \alpha$ for all $i \in [m]$. Now, Corollary 5.1 and Theorem 5.2 imply that we can compute a solution $f_\lambda(\boldsymbol{x}) \leq 2f_\lambda(\boldsymbol{x}^*) + \varepsilon$ after $T = O\left(\lambda^2 X \log \frac{m}{\varepsilon}\right)$ iterations. Now, note that

$$\sum_i \log(1 + e^{-\lambda b_i(\boldsymbol{A}\boldsymbol{x})_i}) \leq 2\sum_i \log(1 + e^{-\lambda b_i(\boldsymbol{A}\boldsymbol{x}^*)_i}) + \varepsilon$$

$$\leq 2m\log(1 + e^{-\lambda\alpha}) + \varepsilon$$

$$\leq 2me^{-\lambda\alpha} + \varepsilon$$

$$\leq 3me^{-\lambda\alpha},$$

after setting $\varepsilon = me^{-\lambda\alpha}$.

Now, re-arranging and using the fact that $3me^{-\lambda\alpha} \leq 2$ implies $e^{3me^{-\lambda\alpha}} \leq 1 + 6me^{-\lambda\alpha}$, we have that

$$b_i(\boldsymbol{A}\boldsymbol{x})_i \geq -\lambda^{-1}\log\left(e^{3me^{-\lambda\alpha}} - 1\right)$$

$$\geq -\lambda^{-1}\log\left(6me^{-\lambda\alpha}\right)$$

$$= \alpha - \lambda^{-1}\log\left(6m\right)$$

$$\geq \alpha(1 - \widehat{\varepsilon}),$$

where the last equality follows by our setting of $\lambda \geq \frac{\log(6m)}{\alpha \widehat{\varepsilon}}$. Therefore, the number of iterations is $O(\lambda^3 \alpha) \leq \widetilde{O}\left(\frac{1}{\alpha^2 \widehat{\varepsilon}^3}\right)$. Additionally, $\|\boldsymbol{x}\|_2 \leq \|\boldsymbol{x}^*\|_2$.

To bound the distance from the classifier, we note that

$$E^2 := \left\|\frac{\boldsymbol{x}}{\|\boldsymbol{x}\|_2} - \boldsymbol{x}^*\right\|_2^2 = \left\|\frac{\boldsymbol{x}}{\|\boldsymbol{x}\|_2} - \boldsymbol{x}^*\right\|_2^2 = 2 - 2\left\langle \frac{\boldsymbol{x}}{\|\boldsymbol{x}\|_2}, \boldsymbol{x}^* \right\rangle.$$

On the other hand, we let $\bar{\boldsymbol{x}} = \frac{1}{2}\frac{\boldsymbol{x}}{\|\boldsymbol{x}\|_2} + \frac{1}{2}\boldsymbol{x}^*$ and compute its smallest margin as

$$\alpha \geq \frac{b_i(\boldsymbol{A}\bar{\boldsymbol{x}})_i}{\|\bar{\boldsymbol{x}}\|_2} \geq \frac{\frac{\alpha(1-\widehat{\varepsilon})}{2\|\boldsymbol{x}\|_2} + \frac{\alpha}{2}}{\sqrt{\frac{1}{4} + \frac{1}{4} + \frac{1}{2}\langle \frac{\boldsymbol{x}}{\|\boldsymbol{x}\|_2}, \boldsymbol{x}^* \rangle}} \geq \frac{\frac{\alpha(1-\widehat{\varepsilon})}{2} + \frac{\alpha}{2}}{\sqrt{1 - E^2/4}}.$$

Re-arranging, we get that

$$1 - E^2/4 \geq \frac{1}{4}(2 - \widehat{\varepsilon})^2 = 1 - \widehat{\varepsilon} + \widehat{\varepsilon}^2/4$$

$$E^2 \leq 4\widehat{\varepsilon}(1 - \widehat{\varepsilon}) \leq 4\widehat{\varepsilon}.$$

Therefore, we have

$$\left\|\frac{\boldsymbol{x}}{\|\boldsymbol{x}\|_2} - \boldsymbol{x}^*\right\|_2 \leq 2\sqrt{\widehat{\varepsilon}},$$

or in other words

$$\left\|\frac{\boldsymbol{x}}{\|\boldsymbol{x}\|_2} - \boldsymbol{x}^*\right\|_2 \leq E$$

after $\widetilde{O}\left(\frac{1}{\alpha^2 E^6}\right)$ iterations. $\qquad\square$

