# OpenReview forum: "Gradient Descent Converges Linearly for Logistic Regression on Separable Data"
_ICLR.cc/2023/Conference — Submitted to ICLR 2023_

### Official Review · Reviewer_9NWd · 2022-10-21

**Confidence:** 4
**Clarity, Quality, Novelty And Reproducibility:** (see above)
**Correctness:** 4
**Technical Novelty And Significance:** 4
**Empirical Novelty And Significance:** 4
**Recommendation:** 8

**Strength And Weaknesses:**

### Strengths

- The paper obtains a new and stronger bound on the convergence of (variable learning rate) gradient descent for logistic regression on separable data. The result is interesting since the loss function of logistic regression is not strongly convex. It also highlights the importance of variable learning rates for gradient descent.
- The mathematical analyses of the work are rigorous and of high quality. Beyond the central result on separable data, several components of the analyses are technically interesting and could be of broad interest, including:
    - the insight about second-order robustness and multiplicative smoothness of logistic regression, which gives rise to an intuitive schedule of the learning rate for better convergence
    - the result on sparse logistic regression, though not completely dominate the result of Shalev-Shwartz et al. (2010), provides an altenative to approach the problem (and is better than the previous bound at least when f(x*) <<m). The result also provides a new technique to analyze convex but not strongly-convex losses for l1-regression.
- The main ideas and techniques of the work were developed based on well-founded rationales. The paper is well-written with a sufficient literature review.

### Weaknesses

- None noted. However, I want to note that the title of the work doesn’t reflect the “variable learning rates” aspect of the result, which is a very important detail. This could lead to the result being misinterpreted in future discussions.

**Summary Of The Paper:**

The paper aims to characterize the convergence of gradient descent (and greedy coordinate descent) with carefully chosen (variable) learning rates for logistic regression. The analysis identifies two special properties of logistic regression loss function: second-order robustness and multiplicative smoothness, which implies that as the loss decreases, the objective function becomes (locally) smoother and the learning rate can be increased (proportional to the value of the loss) to achieve better convergence. This observation is applied to obtain new bounds on the convergence rate for sparse and dense logistic regression, which adds a multiplicative error term to the estimation but only depends logarithmically on the number of sample points. For linearly separable data, the loss approaches zero and the convergence becomes linear and is significantly better than pre-existing bounds.

**Summary Of The Review:**

The paper addresses a meaningful question and the approach is novel and of broad interest. The work is strong in all aspects: novel and practical bounds, rigorous theoretical framework, and useful insights about the theoretical and algorithmic properties of the problem studied.

---

> ### Author Response · Authors · 2022-11-13
> **Response to Reviewer 9NWd**
>
> We thank the reviewer for their positive and detailed review.
>
> Regarding the issue of variable learning rates of gradient descent: We agree that the variable learning rate aspect is important and should not be missed. For this reason, we will change the first sentence of the abstract to: "We show that running gradient descent with variable learning rate guarantees loss [...]"

---

> > ### Comment · Reviewer_9NWd · 2022-12-07
> > **discussions**
> >
> > Dear authors,
> >
> > As you know, we are in the reviewer discussion period, and the reviewers noticed that you haven't uploaded a revision for the manuscript. This makes it more difficult to assess the quality of the paper (specifically in the literature review of the work). I strongly suggest the authors use this feature in response to the reviewers' questions.

---

> > > ### Author Response · Authors · 2022-12-10
> > > **Cannot upload new version**
> > >
> > > Thank you for the comment. Unfortunately we noticed that we are not able to upload a revision any more. We will write a comment summarizing the most significant additions/changes that we have made (especially with regards to literature).

---

### Official Review · Reviewer_qF3Q · 2022-10-24

**Confidence:** 4
**Clarity, Quality, Novelty And Reproducibility:** The paper is well-written, though dis…
**Correctness:** 4
**Technical Novelty And Significance:** 2
**Empirical Novelty And Significance:** Not applicable
**Recommendation:** 5

**Strength And Weaknesses:**

I think linear convergence to $1.1f(x^*)$ is a nice result, since it interpolates nicely between the separable case and non-separable case. It is also interesting to obtain sparsity using coordinate descent. On the other hand, there is a large literature on this topic, which is poorly discussed.
1. [1] considers the exponential loss and learning rate $\frac{1}{f(x_t)\sqrt{t}}$, and shows a $\frac{1}{\sqrt{t}}$ margin maximization rate.
2. Lemma 4.3 of [2] proves the multiplicative smoothness property for the exponential loss and logistic loss.
3. [3] uses an adaptive learning rate and proves $\frac{1}{t}$ margin maximization rate. For the logistic loss, [3] uses learning rate $\frac{1}{2f(x_t)}$ in the first phase and a slightly different normalized learning rate in the second phase. If the maximum margin is denoted by $\alpha$, and the target additive margin-maximization error is $\epsilon$, then [3] needs $\frac{1}{\alpha\epsilon}$ iterations, while this paper needs $\frac{\alpha}{\epsilon^3}$ iterations. Since it only makes sense to let $\epsilon<\alpha$, the convergence rate of [3] is better.

I also have some technical questions:
1. In Theorem 4.1 and 4.3, why do the sparsity bounds depend on $||x^*||_1$? Based on Corollary 4.2, it seems the bounds should depend on $||x^*||_0$.
2. How do you get the $s^4\log^3\frac{1}{\epsilon}$ iteration complexity in Corollary 4.2 from Theorem 4.1?

References:

[1] Mor Shpigel Nacson, Jason Lee, Suriya Gunasekar, Pedro Henrique Pamplona Savarese, Nathan Srebro, and Daniel Soudry, 2019. Convergence of Gradient Descent on Separable Data.

[2] Ziwei Ji and Matus Telgarsky, 2018. Risk and parameter convergence of logistic regression.

[3] Ziwei Ji and Matus Telgarsky, 2021. Characterizing the implicit bias via a primal-dual analysis.

**Summary Of The Paper:**

This paper proposes a greedy coordinate/gradient descent algorithm on logistic regression and proves a multiplicative convergence error and a linear convergence rate. Specifically, for sparse logistic regression, this paper proposes a coordinate descent algorithm where the learning rate is normalized by the logistic loss (or the gradient norm), and shows that the loss converges to $1.1f(x^*)$ at a linear rate where $x^*$ is a reference solution, and also provides a sparsity bound. If there is no sparsity constraint, this gives a convergence rate for dense logistic regression. In the linearly separable case, a poly(1/t) margin maximization rate is proved.

**Summary Of The Review:**

Currently I put the paper marginally below the acceptance threshold; I think a detailed comparison with prior results should be included, and I also hope the authors can answer my questions above.

---

> ### Author Response · Authors · 2022-11-13
> **Response to Reviewer qF3Q**
>
> We thank the reviewer for their detailed review and very helpful literature suggestions. We will update the next version of our paper with a discussion on convergence results for the exponential loss, including the results of [1], [2], [3]. Regarding the multiplicative smoothness results of [2], we will be happy to add a discussion, but we could not directly see a connection between the statement of Lemma 4.3 [2] and multiplicative smoothness. If the reviewer could please clarify whether this is the right lemma to cite, it would be helpful.
>
> Regarding the comparison between the margin maximization results of [3] and our Section 6, we thank the reviewer for bringing this paper to our attention. Indeed it is a very relevant and interesting prior work that we were not aware of, and establishes an $\widetilde{O}\left(1/t\right)$ margin maximization rate. However, there are some subtle but important differences between our margin maximization result and that of [3], which we believe makes them incomparable in the general case. [3] assumes that the data matrix ${\bf A}$ is row $\ell_2$ normalized, while our result doesn't. This is exactly the reason why the bound of [3] doesn't have any dependencies on the conditioning of ${\bf A}$, while ours depends on the multiplicative smoothness of $f$. Row normalization implicitly re-weights the training examples, which is perfectly fine if our goal is margin maximization, but it subtly changes the training loss function. In this sense, our result is somewhat more general, since it shows a slower $O(1/t^{1/3})$ rate but for _any_ logistic loss function on separable data.
>
> But even ignoring the normalization issue, if we assume the empirically observed $\beta m^{-1}$-multiplicative smoothness property from Section 5, our Theorem 6.1 gives $\widetilde{O}\left(\beta m^{-1}\alpha^{-2} \hat{\epsilon}^{-3}\right)$ iterations for $(1-\hat{\epsilon})$-approximate margins. In the case when the constraint matrix is well conditioned (e.g. $\beta = O(\sqrt{m/n})$), this can be considerably lower than the bound $\widetilde{O}\left(\alpha^{-2} \hat{\epsilon}^{-1}\right)$ of [3] for values of $\hat{\epsilon}$ that are not too small. Of course, [3] still obtains better convergence in the worst case, e.g. when the constraint matrix is badly conditioned.
>
> In the next version of the paper, we will add a discussion that acknowledges the $\widetilde{O}(1/t)$ margin maximization results of $[3]$ as well as the above detailed comparison between the two results.
>
> On the technical questions:
>
> 1. The bounds containing the $\ell_1$ norm are actually sharper than those with $\ell_0$. In particular, to obtain Corollary 4.2 from Theorem 4.1, we use the inequality $||x^*||_1 \leq ||x^*||_\infty ||x^*||_0$. We will add this short comment before Corollary 4.2.
>
> 2. The iteration complexity in Corollary 4.2 is
> $\widetilde{O}((||x||_0^2 + ||x^*||_0^2) \log \frac{1}{\epsilon})$.
> Now, $||x^*||_0^2 = s^2$ by definition, and by the sparsity bound of Theorem 4.1,
> $||x||_0^2 \leq \widetilde{O}(||x^*||_1^4 \log^2 \frac{1}{\epsilon}) \leq \widetilde{O}(||x^*||_\infty^4 ||x^*||_0^4 \log^2 \frac{1}{\epsilon} ) \leq \widetilde{O}(s^4 \log^2 \frac{1}{\epsilon})$. So the final bound is $\widetilde{O}(s^4 \log^3 \frac{1}{\epsilon})$. We will add this clarification right before Corollary 4.2.

---

> > ### Comment · Reviewer_qF3Q · 2022-11-20
> > **Wrong claim on prior result**
> >
> > Thanks for your response, but please note that [3] does not require row $\ell_2$ normalization; they only assume the $\ell_2$ norm of inputs are BOUNDED by 1, not equal to 1. The results in [3] can be trivially extended to the general setting by scaling the learning rate.

---

> > > ### Author Response · Authors · 2022-11-26
> > > **Agreed with the comment**
> > >
> > > Thank you for the on point comment. Indeed, we agree that both results work in the general setting with no assumptions on the matrix $A$. A more careful comparison is as follows: The number of iterations in the margin maximization result of [3] has a quadratic dependency on the largest $\ell_1$ row norm of $A$ (this is because of scaling the learning rate) and a linear error dependency. The corresponding result in Section 6 of this paper instead has a linear dependency on the multiplicative smoothness of $A$ (which is related to the squared spectral norm of $A$) but a cubic error dependence. So [3] has a better error dependency, although in general the two bounds are incomparable.

---

> > > > ### Comment · Reviewer_qF3Q · 2022-12-06
> > > > **More clarifications and questions**
> > > >
> > > > Dear authors:
> > > >
> > > > Here are some additional clarifications regarding multiplicative smoothness:
> > > > 1. It is well-known in convex optimization that, given a $\beta$-smooth function $f$, and a gradient descent update $x'=x-\eta\nabla f(x)$, it holds that
> > > > $$f(x')-f(x)\le-\eta||\nabla f(x)||_2^2+\frac{\eta^2\beta}{2}||\nabla f(x)||_2^2.$$ Typically $\eta$ can be chosen to be $1/\beta$, which can guarantee a descent in $f$.
> > > > Now in the case of logistic regression, Lemma 3.4 of [2] proves that $\beta$ can be chosen to be $f(x)$, which will allow for a very large learning rate if $f(x)$ is small. Now it seems to me that Lemma 3.4 of [2] is similar to the proof of Lemma B.1 from this paper, **specifically the descent inequality on $f$ below eq. (8) from this paper**. Can you elaborate the difference between them?
> > > > 2. This property can also be proved via the $\ell_\infty$ smoothness of certain log-sum-exp-style functions: the proof for the exp loss is simpler, but a more general case which involves a "generalized sum" [6] can also be handled (see [3] for a discussion). This had motivated the use of **adaptive learning rates** in a lot of prior work: I have mentioned that [1, 3] used adaptive learning rates to prove fast margin rates for gradient descent. For coordinate descent, the AdaBoost algorithm [7] already used adaptive learning rates, and later [4] proved fast margin rates for CD via adaptive learning rates. [5] analyzed the implicit bias of CD; although they did not use an adaptive learning rate, their analysis also used the idea of multiplicative smoothness.
> > > >
> > > > In the abstract of this paper, it is claimed that " Our key observation is a property of the logistic loss that we call multiplicative smoothness and is (surprisingly) little-explored: As the loss decreases, the objective becomes (locally) smoother and therefore the learning rate can increase." Because of the literature I mentioned above (and many others I didn't mention), I can't agree this topic is "little explored".
> > > >
> > > > Finally, on the margin maximization rate of [3], please note that it depends on the largest **l2** row norm of $A$, not l1 row norm. In your latest response, you also mentioned that your convergence rate depends on the spectral norm of $A$; however, I couldn't find any discussion on the spectral norm in the current submission. Can you elaborate what is the exact dependency on the spectral norm of $A$? Specifically, the rate in Theorem 6.1 of this paper depends on $X$, and right above Theorem 6.1, it is mentioned that $X$ "depends on whether we use Corollary 5.1 or Theorem 5.2, but is beyond the point of this section, since here we are interested in the error dependence." I would like to know what **exactly** $X$ is.
> > > >
> > > > In fact, Corollary 5.1 of this paper depends on $n^2M^2$, where $n$ is the input dimension and $M$ is the $\ell_\infty$ norm of the inputs, so this is worse than a requirement on the $\ell_2$ norm of the inputs. Theorem 5.2 also depends on $R$, which is basically the l2 norm of the optimal solution. It is unclear to me how to derive a bound on $X$ from Corollary 5.1 and Theorem 5.2, but I believe it is important.
> > > >
> > > > [4] Matus Telgarsky, 2013. Margins, shrinkage, and boosting.
> > > >
> > > > [5] Suriya Gunasekar, Jason Lee, Daniel Soudry, and Nathan Srebro, 2018. Characterizing implicit bias in terms of optimization geometry.
> > > >
> > > > [6] G. H. Hardy, J. E. Littlewood, and G. Polya, 1934. Inequalities.
> > > >
> > > > [7] Yoav Freund and Robert E. Schapire, 1997. A decision-theoretic generalization of on-line learning and an
> > > > application to boosting.

---

> > > > > ### Author Response · Authors · 2022-12-10
> > > > > **Response to clarifications**
> > > > >
> > > > > We thank the reviewer for their very detailed comments.
> > > > >
> > > > > 1. Indeed the proof of Lemma 3.4 of [2] and the proof of Theorem B.1 both use a second-order Taylor expansion combined with the property $\ell'' \leq \ell$ of the logistic loss.
> > > > > However, there are some important differences:
> > > > >
> > > > >    - The biggest technical difference in our opinion is that Lemma 3.4 of [2] uses the first-order property $\ell' \leq \ell$ (essentially the fact that it doesn't grow faster than an exponential), while Theorem B.1 of this paper uses the second-order robustness property, which is related to the third-order condition $\ell''' \leq \ell''$. This is why our step size is the minimum of two branches, and not proportional to $1/f(x)$. As such, the two results apply to different sets of loss functions, but the logistic loss and exponential losses lie at the intersection of these sets. In fact, the proofs are quite different and interesting in their own right, and after reading Lemma B.2 of [2] there are some points that we still can't say we fully understand (specifically, the relationship between the property $\ell' \leq \ell$ and the risk inequality $\left\|\nabla \mathcal{R}(w)\right\|^2 \leq \mathcal{R}(w)^2$).
> > > > >
> > > > >    - Another difference has to do with how this Lemma is ultimately used, which has to do with the definition of the multiplicative smoothness property and how the structure of the data matrix of $A$ enters the proof (and bounds). While [2] uses the maximum row $\ell_2$ norm of $A$, the results of Section 5 (Dense logistic regression) use the spectral norm $\beta$ of $A$, together with the experimentally observed assumption in Table 3.
> > > > >
> > > > > 2. Thank you for bringing these papers to our attention. We agree that, based on the literature that you have pointed out, similar properties have been used before and
> > > > > the statement on the obscurity of the multiplicative smoothness property is too strong. We will remove this statement from the abstract and revise it to reflect that part of our contribution is on how we are using it,
> > > > > in combination with the second-order robustness property.
> > > > >
> > > > > 3. We happily elaborate and will include the following discussion in the paper, while removing any reference to $X$. $X$ can be bounded by $\mu + \gamma$, where $\mu$ is the multiplicative smoothness constant and $\gamma$ is the second-order robustness constant of $f$. (If we want to be exact, the number of iterations is bounded by $\widetilde{O}\left(\mu \alpha^{-2} \hat{\epsilon}^{-3} + \gamma \alpha^{-1} \hat{\epsilon}^{-2}\right)$).
> > > > > In the case when $f$ is the logistic loss with data matrix $A$, $\gamma$ can be bounded by $2M$, where $M$ is a bound on the entries of $A$ (in absolute value). We can further bound this by $2\sqrt{\beta}$, where $\beta$ is the spectral norm of $A^{\top} A$.
> > > > > Now, for $\mu$, if we use the inequality right after Corollary 5.1, we get $\mu \leq \beta$ leading to $X \leq O(\beta)$ and an iteration bound of
> > > > > $\widetilde{O}\left(\beta \alpha^{-2} \hat{\epsilon}^{-3}\right)$
> > > > > (which is always worse than the $\max_i ||A^\top 1_i||_2^2 \alpha^{-2} \hat{\epsilon}^{-1}$ bound of [3]).
> > > > > If we instead use the multiplicative smoothness assumption that is described in Section 5 (Specifically that $\langle w(x), (A \nabla f(x))^2\rangle \leq f(x) m^{-1} ||A\nabla f(x)||_2^2$), together with the definition of the spectral norm $||A\nabla f(x)||_2^2 \leq \beta ||\nabla f(x)||_2^2$, we get $\mu \leq \beta m^{-1}$. This implies a bound $X\leq O(\beta m^{-1} + \sqrt{\beta})$ for a final iteration bound of $\widetilde{O}\left(\beta m^{-1} \alpha^{-2} \hat{\epsilon}^{-3} + \sqrt{\beta} \alpha^{-1} \hat{\epsilon}^{-2}\right)$.
> > > > >
> > > > > The reviewer is right that $R$ is the $\ell_2$ norm of the optimal solution and there is an $R^2$ dependence. However, the constrained optimization problem which we are solving is $\min_x f(x)$, under $||x||_2 \leq p$ (This is in fact equivalent to optimizing $\min_x f_p(x)$ under $||x||_2 \leq 1$, and for this reason in our new version of the paper we simplified it so that there is no $f_p$). Based on the analysis in Theorem 6.1, we can set $p = \alpha^{-1} \hat{\epsilon}^{-1} \log(6m)$, and so $R^2 = p^2 = \widetilde{O}\left(\alpha^{-2} \hat{\epsilon}^{-2} \right)$, which is exactly where this runtime dependency we mentioned before is coming from. The rest of the dependency is coming from $\log \frac{1}{\epsilon} = \widetilde{O}\left(p\alpha\right) = \widetilde{O}\left(\hat{\epsilon}^{-1}\right)$, leading to
> > > > > the bound if $\alpha^{-2} \hat{\epsilon}^{-3}$.
> > > > >
> > > > > Please let us know if there is something else that we can clarify and we will be happy to.

---

> > > > > > ### Comment · Reviewer_qF3Q · 2022-12-10
> > > > > > **Further response**
> > > > > >
> > > > > > Thanks for the response.
> > > > > >
> > > > > > 1. Regarding Lemma 3.4 of [2] and Lemma B.1 of this paper, I agree the analyses are not exactly the same. On the other hand, as the authors mentioned, both [2] and this paper use the property $\ell^{''}\le\ell$, which directly motivates the multiplicative smoothness property. In fact, under eq. (3) from this paper, this property is described as "the main observation on which our analysis is based", and thus I think it should be emphasized that prior work had similar observations.
> > > > > > 2. In fact, the effective multiplicative smoothness property in Section 5 of this paper focuses on the quantity $\langle w(x), (A\nabla f(x))^2\rangle$, and the **same** quantity is analyzed in the proof of Lemma B.1 of [2]. It is also related to the discussion of matrix norms used in different analyses: at the bottom of page 7 of this paper, there is an inequality chain which includes $||A||_\{2\to\infty}^2$. This norm is naturally bounded by the square of largest row $\ell_2$ norm of $A$, which is used in the proof of Lemma B.1 of [2]. On the other hand, this paper tries to bound $||A||_\{2\to\infty}^2$ by $\beta$, the squared spectral norm of $A$ (sorry I didn't catch this previously), but this might be too loose since the spectral norm depends on $m$, the number of rows of $A$. This paper further claims that $\beta$ can be improved to $\beta/m$ based on empirical observations; unfortunately this is **not a rigorous proof**, and when these empirical observations hold, of course prior analyses can also be improved.
> > > > > > 3. Regarding the inequality $||\nabla\mathcal{R}(w)||_2^2\le\mathcal{R}(w)^2$ from [2], as far as I understand, it comes from $\ell'\le\ell$, the condition (without loss of generality) that $||x_i||_2\le1$, and the triangle inequality.

---

> > > > > > > ### Author Response · Authors · 2022-12-10
> > > > > > > **Response**
> > > > > > >
> > > > > > > Thank you very much for all the comments. We will be happy to emphasize all these observations in our next revision.

---

### Official Review · Reviewer_gbUi · 2022-10-24

**Confidence:** 4
**Correctness:** 4
**Technical Novelty And Significance:** 3
**Empirical Novelty And Significance:** 3
**Recommendation:** 8

**Clarity, Quality, Novelty And Reproducibility:**

Overall, I find the paper is clearly written. The quality of the theoretical results is high. The key observation, which is the multiplicative smoothness, seems to be novel; and the linear convergence rate results appear to be novel. In terms of reproducibility, I believe that the proofs should be correct, though I didn't check them very carefully.

**Strength And Weaknesses:**

This paper follows an interesting line of work where people study the implicit bias of GD algorithms for separable data (or more generally, data under over-parametrized models). The algorithmic aspect of implicit bias is the convergence properties of GD, including the convergence rate in particular. It is known that, somewhat undesirably, the convergence is very slow for logistic loss.

The idea of adapting the step size to objective value at each iteration is quite natural. I am not sure if multiplicative smoothness or similar conditions have been studied before, but it appears to me that this critical condition captures the structure of logistic loss well, and it is key to establishing the linear convergence rate. As far as I know, this multiplicative smoothness condition is a novel and useful notation.

In my opinion, the novelty of the multiplicative smoothness condition, the generality of the setup (sparse linear regression), and the strong results (linear convergence), are the main strengths of this paper.

One weakness is that the empirical condition on multiplicative smoothness assumed in Theorem 5.2 needs more justification. I imagine that
one can perhaps use some random matrix theory to provide an argument for why the empirical multiplicative smoothness is smaller by a factor of $m$. For example, if the matrix $A$ is random and $A, x$ are independent, then the trivial bound on multiplicative smoothness is too pessimistic and thus can be improved. Perhaps it can be shown empirically that even running a few iterations $A, x$ remain weakly dependent so multiplicative smoothness can still be bounded well. See [] for instance.

There are a few other places that need clarification.
- There are some statements involving "for any solution $x^*$". For example, in Corollary 5.1, it is assumed that $x^*$ lies in $[-B,B]^n$. Do we choose $x^*$ to be the max-margin solution, or a scalar times the max-margin solution? Or is it an arbitrary deterministic solution?
This needs to be stated more clearly because the result depends on $B$.
- The connection between coordinate descent and gradient descent is not clear. So it is a bit confusing how the results in Section 5 are related.


[2] Y Zhong, N Boumal, Near-optimal bounds for phase synchronization, SIAM Journal on Optimization, 2018
[3] Y Chen, Y Chi, J Fan, C Ma, Gradient descent with random initialization: Fast global convergence for nonconvex phase retrieval, Mathematical Programming, 2019

**Summary Of The Paper:**

In this paper, the authors studied logistic regression on separable data. It is known, for example in [1], that when the data is linearly separable, gradient descent (GD) on the logistic loss converges to the max-margin classifier with a very slow rate $O(\log \log t/\log t)$. In the setting of sparse linear regression, the author considered coordinate descent and GD with varying step sizes---which builds on ideas from prior papers. A key contribution is the observation that logistic loss satisfies a condition called multiplicative smoothness, under which the step size is allowed to increase, which speeds up the convergence. Under the multiplicative smoothness condition, linear convergence results are established.

[1] Daniel Soudry, Elad Hoffer, Mor Shpigel Nacson, Suriya Gunasekar, and Nathan Srebro, The im- plicit bias of gradient descent on separable data. The Journal of Machine Learning Research, 2019


**Summary Of The Review:**

This paper is a nice addition to the growing literature on implicit bias and convergence properties of GD. I would recommend acceptance of this paper.

---

> ### Author Response · Authors · 2022-11-13
> **Response to Reviewer gbUi**
>
> - On theoretical grounding for the improved multiplicative smoothness assumption: We thank the reviewer for the thoughtful suggestions and pointing out the related works. We think this is a non-trivial and interesting question, and something that we plan to further pursue in future work. We have some early stage ideas but it still requires a significant amount of work to lead to a potential result.
>
> - On the definition of $x^*$: We thank the reviewer for pointing this out. In all our statements (except for Theorem 6.1 where $x^*$ is explicitly defined as the maximum margin solution), $x^*$ is any arbitrary fixed solution in $[-B,B]^n$. For results that are concerned with bounding the loss value, one could alternatively define $x^*$ as a minimizer of $f$ inside the box $[-B,B]^n$. We will add this short discussion in the abstract, after Theorem 4.1, and before Theorem 6.1 in order to avoid any ambiguity.
>
> - On the connection between coordinate descent and gradient descent: Thank you for the question, we will add a paragraph in the manuscript to clarify this connection, and please feel free to ask for additional clarifications. Gradient descent for general norms (aka steepest descent) takes a step in the direction $\delta$ that minimizes the quantity $\langle\nabla f(x), \delta\rangle + \frac{1}{2} ||\delta||^2$. If $||\cdot||$ is the $\ell_2$ norm, we get gradient descent, where $\delta = -\nabla f(x)$. If $||\cdot||$ is the $\ell_1$ norm, then $\delta$ is a greedy coordinate descent step, i.e. $\delta_i = -\nabla_i f(x)$ where $i$ is a coordinate that maximizes $|\nabla_i f(x)|$. We could use either of these optimization algorithms. Greedy coordinate descent is suitable for sparse optimization, because it only updates one coordinate at a time. On the other hand gradient descent generally performs better in terms of convergence in practice, but gives no guarantees on the sparsity of the solution. Interestingly, even if we don't care about sparsity, the worst case analysis in Corollary 5.1 still gets its best possible bounds by using a greedy coordinate descent step. We then propose Theorem 5.2 to bridge this mismatch between the worst case analysis, and the practical results which show that gradient descent converges much faster than greedy coordinate descent. In practical terms, the difference between Corollary 5.1 and Theorem 5.2 is that the former is a worst case analysis that only assumes a bound on the magnitude of entries of $\bf A$, while the latter obtains much better convergence while assuming a stronger (but empirically validated) multiplicative smoothness condition.

---

### Official Review · Reviewer_DG4g · 2022-10-28

**Confidence:** 3
**Clarity, Quality, Novelty And Reproducibility:** This paper is well written. See detai…
**Correctness:** 3
**Technical Novelty And Significance:** 3
**Empirical Novelty And Significance:** 2
**Recommendation:** 6

**Strength And Weaknesses:**

This paper exploits two interesting properties of the logistic loss function, namely second-order robustness and manipulatively smoothness, to prove the sublevel set linear convergence results. The improved sparsity and alignment also seem very valuable for the community. My concerns are as follows:

* The linear convergence results only concern the computation of a point with a sublevel set, which is fundamentally different from the sublinear rate of gradient descent methods. It might be somehow misleading to use the notation x^* in Tables 1 and 2 as for any bounded x^*, GD also has a linear convergence rate (as the loss function is sharp around such x^*). The "Guarantee" of the sublinear rate of GD should be written with inf f. It is recommended to clarify the meaning of x^* early in the introduction, as this might cause confusion even in the abstract part.

* On Thm 5.2: it seems this theorem is a generalized version for any convex function with second-order robustness and multiplicative smoothness. But if we have f(x^*) < 0 and f(x^0) > 0, the statement seems problematic. We can choose a sufficient large delta such that (1+delta) f(x^*) + epsilon < inf f. The problem seems from Lemma B.1, in which the function is required to have a positive range.

Minor:
* Eq.(2): "=" should be "<="

**Summary Of The Paper:**

This paper considers the optimization of the logistic regression problem with separable data assumption. They observe that when the iteration variable is far from zero, the smoothness parameter decreases (i.e., the function is smoother), which allows more aggressively long step size. They prove the linear convergence of the modified algorithm to a multiplicative sublevel set. They claim this result improves the dependency on the magnitude of the optimal solution exponentially compared with existing works. They also show rate improvement on the sparsity and speed of alignment to a maximum margin estimator.

**Summary Of The Review:**

In sum, this paper studies a classic ML model and introduces interesting new analytic and algorithmic techniques to prove linear convergence to a sublevel set with better dependency on the magnitude of the optimal solution. The improved sparsity and alignment also seem very valuable for the community.

---

> ### Author Response · Authors · 2022-11-13
> **Response to Reviewer DG4g**
>
> We thank the reviewer for the helpful suggestions.
>
> - We will add the clarification about $x^*$ being an arbitrary solution and not a global optimum in the abstract, after Theorem 4.1, and before Theorem 6.1. Regarding the convergence guarantees in Table 1 (and 2), we agree that the presentation can be improved because gradient descent achieves linear rate in any bounded set, but with an exponential dependency in the entries of $x^*$. In order to resolve this, we will change the captions to additionally state that polynomial dependencies in the entries of $x^*$ are omitted, and also clarify that the standard linear convergence analysis gradient descent has an _exponential_ dependence on the entries of $x^*$, thus it is not included in the table.
>
> - On Thm 5.2: Thank you for the careful observation, indeed this is a typo, the function is required to have a positive range.

---

### Author Response · Authors · 2022-12-10
**Summary of updates**

We again thank all the reviewers for their detailed comments. We have thought carefully about all their suggestions and incorporated them in the manuscript. As we are not able to upload a manuscript revision, we present the most important changes in the abstract and literature review (intro) below.

**Abstract**
> We show that running gradient descent with variable learning rate guarantees loss
$f(x) \leq 1.1 \cdot f(x^*) + \epsilon$
for the logistic regression objective,
where the error $\epsilon$ decays exponentially with the number of iterations and polynomially with the magnitude of the entries of
an arbitrary fixed solution $x^*$.
This is in contrast to the common intuition that the absence of strong convexity precludes linear
convergence of first-order methods, and highlights the importance of variable learning rates
for gradient descent.
Our results also directly imply margin maximization results for separable data, specifically on
the convergence of the classifier returned by gradient descent to the hard SVM classifier.
We also apply our ideas to sparse logistic regression, where they lead to an exponential improvement of the sparsity-error tradeoff using greedy coordinate descent.


**Introduction**
> Logistic regression is one of the most widely used classification methods because of its simplicity, interpretability, and good practical performance. Yet, the convergence behavior of first-order methods
on this task is not well understood: In practice gradient descent performs much better than what the theory predicts.
In particular, a general analysis of gradient descent for smooth functions implies convergence with the error in function value decaying as $O(1/T)$.
Analyses with stronger, linear convergence guarantees generally require the function to satisfy the strong convexity property, which, in contrast
to other losses such as the $\ell_2$ loss, the logistic loss only satisfies in a bounded set of solutions around zero.
As a result, this introduces an _exponential_ runtime dependency on the magnitude of the
target solution Ratsch et al. (2001); Freund et al. (2018), which is undesirable in practice.
This poses a serious obstacle to obtaining high-precision solutions for logistic regression.

> In fact, it was shown in Telgarsky & Singer (2012)
that the $\mathrm{poly}(1/T)$ bound on function value convergence is tight for gradient descent on
general (non-linearly separable) data.
The significance of the separability of the data for convergence has also been observed
in Telgarsky (2013); Ji & Telgarsky (2018); Freund et al. (2018), who present
convergence results based on quantitative measures of separability.

> A deeper study into the structure of both the exponential and logistic losses
for separable data was initiated
by Telgarsky & Singer (2012), who showed that greedy coordinate descent
achieves linear convergence with a rate that depends on the maximum linear classification margin
(i.e. hard SVM margin). Unfortunately, for logistic regression, it also has a $2^m$ dependence on the
number of examples, making it inefficient for real-world tasks.
Telgarsky (2013) refines the results of Telgarsky & Singer (2012) for the exponential loss,
but for logistic regression still suffers from an exponential overhead originating
from the multiplicative discrepancy between the exponential and logistic losses.
Interestingly, however, the authors note (Telgarsky (2013), Section 5)
that logistic regression
experiments paint a much more favorable picture than the theory predicts.

> A related line of work deals with convergence to the maximum-margin classifier on linearly
separable classification instances using gradient descent.
Soudry et al. (2018); Ji & Telgarsky (2018) showed that the estimator obtained by optimizing the logistic
or the exponential loss with gradient descent
converges to the maximum-margin linear classifier at a rate of
$O(\log \log T / \log T)$ (in $\ell_2$ norm).
For the exponential loss, Nacson et al. (2019) showed that
the convergence bound to the maximum margin estimator can be exponentially improved to
$O(\log T / \sqrt{T})$, by using gradient descent with variable (increasing) learning rate.
The authors' experiments indicate that variable step sizes could lead to a similar exponential
improvements for the case of logistic regression and shallow neural networks.
Recently, Ji & Telgarsky (2021) presented a novel primal-dual approach that proves
that the latter claim indeed holds for the logistic regression and exponential objectives,
obtaining a maximum-margin error decaying as $O(1/T)$, using a variable learning rate. This exponentially
improved upon the results of Soudry et al. (2018); Ji & Telgarsky (2018).

---

### Decision · Program_Chairs · 2023-01-20

**Decision:**

Reject

**Justification For Why Not Higher Score:**

Significant issues related to prior work.

**Justification For Why Not Lower Score:**

N/A

**Metareview: Summary, Strengths And Weaknesses:**

This paper studies greedy coordinate descent and gradient descent on the logistic loss, in both separable and nonseparable scenarios (a more detailed summary is included below in this meta-review).  The paper has many strengths, however ultimately there was concern over missing references, including references containing claimed contributions of this work.  The authors did not upload a revision, and substantial changes are required, given the extent of the reference updates needed (see, for instance, below, and in reviewer comments).  As such, it is suggested that the authors continue and further revise their work, with careful attention to prior work, and submit to a later venue.

Detailed comments are as follows.

This paper studies convergence rates of greedy coordinate descent and gradient descent on logistic regression.  The work is reliant upon the fact that exponentially-tailed losses maintain a relationship between smoothness constants and function values; this relationship dates back at least to AdaBoost, and has been exploited for steepest descent with all lp norms.  Though this work cites some of these papers, it mistakenly refers to this smoothness relationship as a new discovery; see, e.g., the reviewer discussion below for many further references.  The other component of the paper are "oracle" style bounds that pay for the norm of a comparator.  This proof technique can be found in many places, for instance most closely in the paper "boosting with early stopping: [...]" by Zhang and Yu.

Further comments:

- A key issue with the paper, identified in the reviews below, are missing references and incomplete discussion of referenced papers for multiplicative smoothness and margin maximization rates.  The authors chose not to revise the paper in openreview, and later suggested changes to the first pages in an openreview comment, however it appears that much more extensive changes are needed.

- In the case of coordinate descent, the paper refers to work of Shalev-Shwartz and Singer, but that work uses Franke-Wolfe variants, whereas works of Telgarsky referenced by the reviewers directly invoke coordinate descent and are more similar (and also contain a discussion comparing to Shalev-Shwartz and Singer).

Minor comments:

- The presentation of coordinate descent has some modifications to the step size, specifically the function zeta.  Why is zeta there?  That is not present in AdaBoost (which uses "multiplicative smoothness" and achieves exponential rates) and does not seem necessary.  Step size care does appear in the work of Zhang-Yu referenced above, but overall this is a bit of a delicate discussion and should not be omitted.